# AURA: Structural and Semantic Calibration for Robust Federated Graph Learning

## Abstract

Training highly generalizable server model necessitates requires data from multiple sources in Federated Graph Learning. However, noisy labels are increasingly undermining federated system due to the propagation of erroneous information between nodes. Compounding this issue, significant variations in data distribution among clients make noise node detection more challenging. In our work, we propose an effective structural and semantic calibration framework for Robust Federated Graph Learning, `AURA`. We observe that spectral discrepancies across different clients adversely affect noise detection. To address this, we employs SVD for self-supervision, compelling the model to learn an intrinsic and consistent structural representation of the data, thereby effectively attenuating local high-frequency perturbations induced by noisy nodes. We introduce two metrics, namely "Depth Influence" and "Breadth Influence". Based on these metrics, the framework judiciously selects and aggregates the most consensual knowledge from the class prototypes uploaded by each client. Concurrently, clients perform knowledge distillation by minimizing the KL divergence between their local model's output distribution and that of the global model, which markedly enhances the model's generalization performance and convergence stability in heterogeneous data environments. `AURA` demonstrates remarkable robustness across multiple datasets, for instance, achieving a $7.6\% \uparrow$ F1-macro score under a 20%-uniform noise on Cora. The code is available for anonymous access at https://anonymous.4open.science/r/AURA-F351/.

## 1 Introduction

Federated Graph Learning (FGL), which combines the advantages of federated learning and graph neural networks, is rapidly emerging as a significant research direction in distributed machine learning. This approach enables multiple clients to collaboratively train a shared global model while preserving the privacy of local graph data. It ensures effective privacy protection (13; 14) and utilizes the neural message passing mechanism to learn representations from non-Euclidean data structures. Such graph-structured data is prevalent in domains including biological networks (45), urban transportation systems (44), and online social platforms (33).

Despite its potential, existing FGL research often relies on the idealized assumption of abundant and accurate node labels. In practice, however, obtaining large-scale, high-quality labeled datasets is a costly endeavor. Consequently, methods such as web scraping or utilizing user-generated content for automatic label acquisition have become common. While these approaches reduce costs, they inevitably introduce challenges related to label sparsity and noise.

It is well-established that neural networks are prone to overfitting noisy labels, which degrades their generalization performance (28; 43). As a generalization of neural networks to the graph domain, FGL models, as illustrated in Figure 1, exhibit similar performance degradation when trained with noisy labels. A more severe issue arises from the inherent message-passing mechanism of GNNs, which exacerbates the negative impact of label noise. Unlike independently and identically distributed (i.i.d.) data, where a label error is isolated, an incorrect label on a single node can propagate to its neighbors via graph edges. This process triggers cascading errors throughout the multi-layer propagation of GNNs. This pollution propagation effect ultimately leads to a significant degradation in the global model's performance, far exceeding the impact observed on i.i.d. data.

The complexity of this problem is further amplified within the federated learning framework, where the unique graph topology of each client introduces significant heterogeneity.

From a structural perspective, node labels are not isolated attributes but are closely coupled with their local connectivity patterns. A clean label typically aligns with its neighborhood, manifesting as a smooth, low-frequency signal in the graph's spectral domain. In contrast, a noisy label disrupts this local consistency, appearing as an abrupt, high-frequency anomaly. Significant structural differences among client graphs also lead to varied spectral compositions. For instance, in graphs with high homophily (*e.g.*, social networks), noisy nodes generate more discernible high-frequency signals. Conversely, in graphs with complex connectivity patterns (*e.g.*, biological molecular networks), these local anomalies can be easily obscured by the inherent structural noise. Consequently, noise detection methods effective in one spectral environment may fail in another. Thus, a key question arises: 1) *How can we design a framework that generalizes across heterogeneous client graphs by mining intrinsic structural patterns associated with label quality, to distinguish between clean and noisy nodes and mitigate performance degradation from client drift?*

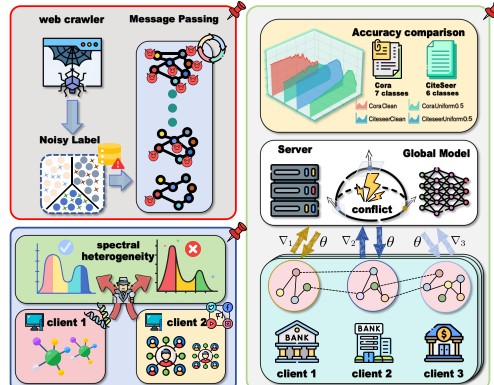

Figure 1: **Problem Illustration**. For current robust FGL scenarios: **I)** As illustrated in the top-right subplot, noisy labels have a severely detrimental effect on the server model's performance, causing a performance drop of nearly 50%. **II)** The heterogeneity of data distributions across clients poses a significant challenge to the detection of noisy nodes.

From a semantic perspective, even if clients can learn representations that are more robust to local spectral and noise variations, the server still faces significant aggregation conflicts during the knowledge fusion stage. Due to spectral heterogeneity and the non-i.i.d. nature of the data, updates from different clients naturally carry different or even contradictory semantic directions. Consequently, the server receives multiple, potentially conflicting updates for the same conceptual class. If a simple averaging strategy like FedAvg is used, these differences are often blurred, leading to a global consensus that is either suboptimal or deviates from the correct direction. This naive aggregation approach not only fails to extract shared knowledge effectively but can also misguide the local models in subsequent rounds. Therefore, a more fundamental question is: 2) *How can we filter out local noise and drift while constructing a high-quality global knowledge base to guide all clients?*

To address the client drift issue arising from spectral heterogeneity, as identified in Problem I, we propose Structural-aware Frequency Alignment (SFA). This method utilizes Singular Value Decomposition (SVD) to perform a low-rank approximation of each client's adjacency matrix. This process decomposes the original graph into two distinct views: a "structural backbone view" capturing core, low-frequency structural information, and a "redundant view" containing high-frequency and noise-related information. By enforcing representational consistency across these two views, our model performs dual denoising, simultaneously addressing inter-client heterogeneity and intra-client label noise. This approach effectively filters out high-frequency variations arising from structural differences among clients, thereby mitigating client drift. Moreover, this framework concurrently weakens the impact of local high-frequency disturbances caused by mislabeled nodes. The resulting node embeddings are highly robust to both noise and drift, providing a solid foundation for downstream tasks.

To address the semantic aggregation conflicts in Problem II, we propose Semantic-guided Consensus Distillation (SCD). The core of SCD is constructing a stable, high-quality global prototype graph to serve as a "semantic anchor" for alignment across all clients. On the client side, we first define depth influence and breadth influence, then select and aggregate local key information based on these metrics before uploading it to the global model. On the server side, a similarity-based selection mechanism aggregates the most consensual knowledge from client-uploaded category prototypes to build this global graph. Clients continue to employ relational knowledge distillation. By minimizing the Kullback-Leibler divergence between the similarity distributions of local and global model outputs with respect to the global prototypes, we compel the local model to emulate the global model's understanding of the semantic relationships within the prototype graph. This approach elevates knowledge transfer from a hard, vector-level alignment to a soft, semantic-relational

one, significantly enhancing generalization and stability. This framework accurately decouples label noise and interference from spectral heterogeneity while ensuring excellent performance. We name the combination of these two strategies `AURA`, the Structural and Semantic Calibration framework for Robust Federated Graph Learning. Our main contributions are summarized as follows:

- We address the challenging problem of defending against noisy labels in FGL. We identify the key challenges as learning representations robust to the dual interference of intra-client label noise and inter-client spectral heterogeneity, and constructing a high-quality global knowledge base while mitigating the effects of local noise and client drift.

- We propose `AURA`, a novel framework that combats noisy labels in FGL via a dual strategy of Representation Purification and Knowledge Alignment.

- We conduct extensive experiments on five mainstream datasets under various noise types and ratios. The results demonstrate that our approach significantly outperforms existing state-of-the-art FGL methods. For instance, under the 20% uniform noise setting on the Cora dataset, our method outperforms the second-best method by a significant margin of 7.6% in F1-macro score.

## 2 PRELIMINARIES

**Notations.** We follow the general paradigm of federated graph learning, $K$ participants (indexed by k) collaboratively train a shared global model using their private graph data. Participant $k$ holds a graph $\mathcal{G}_k = (\mathcal{V}_k, \mathcal{A}_k, \mathcal{X}_k)$, where $\mathcal{V}_k = \{v_i\}_{i=1}^{N_k}$ is the node set containing $|\mathcal{V}_k| = N_k$ nodes, $\mathcal{A}_k \in \{0,1\}^{N_k \times N_k}$ is the adjacency matrix with $A_{ij} = 1$ if there is an edge between nodes $v_i$ and $v_j$ (and 0 otherwise). Similarly, $\mathbf{X}_k$ represents node features, and $\mathbf{Y}_k$ represents the corresponding label set.

**Problem Formulation.** To evaluate the robustness of our method, we take the semi-supervised node classification on the graph as the pretext task which can be defined as follows. We split all the nodes $\mathcal{V}$ into three sets $\mathcal{V}^{train}$, $\mathcal{V}^{val}$ and $\mathcal{V}^{test}$ for training, validation, and testing respectively. After label contamination, $\tilde{\mathbf{Y}}$ is divided into the ground truth labels $\mathcal{Y}$ and the corrupted labels $\tilde{\mathcal{Y}}$. When learning representations in noisy label scenarios, clean nodes and noisy nodes in $\mathcal{V}^{train}$ are available. Besides, given $\mathcal{X}_k$ and $\mathcal{A}_k$, the goal of node classification is to train a classifier $Enc_{\theta_k} : (\mathcal{X}_k, \mathcal{A}_k) \to \hat{\mathcal{Y}}_k$, where the model parameters are optimized by minimizing the following objective:

$$\min_{\theta_k} \mathcal{L}(Enc_{\theta_k}(\mathcal{X}_k, \mathcal{A}_k), \mathcal{Y}_k), \tag{1}$$

where $\mathcal{L}$ is a loss function that quantifies the discrepancy between predictions and ground-truth labels. In accordance with the principle of Empirical Risk Minimization (ERM), the well-trained classifier $Enc_{\theta_k}$ is capable of generalizing well to unseen nodes in $\mathcal{V}^{test}$.

**Uniform noise (34):** This noise model assumes that the true label has a probability $\in (0,1)$ of being uniformly flipped to any of the other classes with equal probability. Formally, for all $j \neq i$,

$$p(y_m^N = j \mid y_m^L = i) = \frac{\epsilon}{d-1}. \tag{2}$$

**Pair noise (41):** This noise model assumes that the true label can only be flipped to a specific paired class with a fixed probability $\epsilon$, while remaining unchanged with probability $1 - \epsilon$.

The optimization objective is to learn a generalizable global model through the federated learning process that performs well under noisy conditions while maintaining strong robustness.

## 3 METHODOLOGY

### 3.1 STRUCTURAL-AWARE FREQUENCY ALIGNMENT (SFA)

**Motivation.** Significant structural differences among client graphs lead to varied spectral compositions. Consequently, a noise detection method effective in one spectral environment may prove ineffective in another. To design a noise filtering framework that is insensitive to these spectral variations, we propose Structural-aware Frequency Alignment (SFA). This method leverages Singular Value Decomposition (SVD) to perform a low-rank approximation of each client's adjacency matrix, simultaneously addressing both inter-client heterogeneity and intra-client label noise. **Graph Spectral Structural Decomposition.** To create the structural backbone view, we perform a low-rank

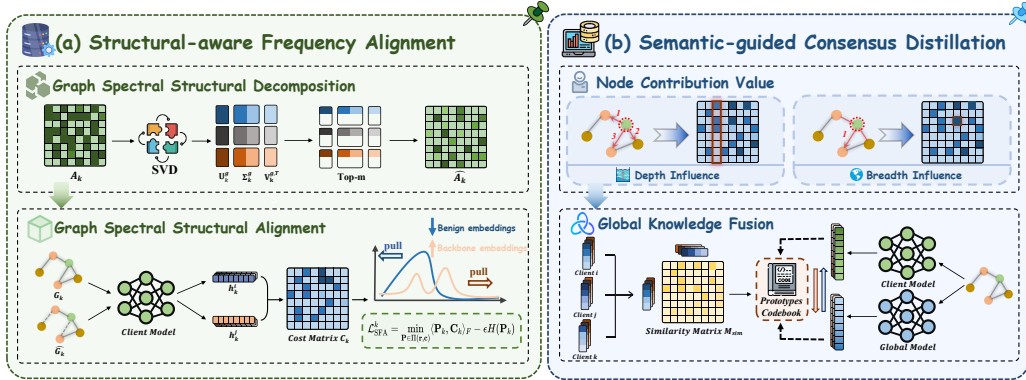

Figure 2: **Architecture illustration** of `AURA`. Best viewed in color and zoom in for details.

approximation on each client's adjacency matrix $\mathbf{A}_k$. First, we apply Singular Value Decomposition (SVD) to decompose $\mathbf{A}_k$ into its constituent structural components:

$$\mathbf{A}_k = \mathbf{U}_k \boldsymbol{\Sigma}_k \mathbf{V}_k^T, \tag{3}$$

where $\mathbf{U}_k$ and $\mathbf{V}_k^T$ are orthogonal matrices of left and right singular vectors, and $\boldsymbol{\Sigma}_k$ is a diagonal matrix of singular values. The singular values in $\boldsymbol{\Sigma}_k$ represent the importance of their corresponding structural patterns. In particular, the largest singular values in $\boldsymbol{\Sigma}_k$ and their corresponding vectors capture the most dominant, low-frequency structural patterns.

Multiple articles ([8]; [6]) demonstrate that these core structural patterns are more critical for learning robust node representations than high-frequency details or noise. We posit that these patterns form the graph's structural backbone and are more critical for learning robust representations. Conversely, the smaller singular values correspond to high-frequency details, which can include both client-specific structural heterogeneity and noise from sources like incorrect labels.

Therefore, we construct the low-frequency backbone view by retaining only the top-m singular values and their corresponding vectors, effectively capturing the essential structural information. Let $\mathbf{U}_k^{[1:m]}$, $\boldsymbol{\Sigma}_k^{[1:m]}$ and $\mathbf{V}_k^{[1:m]}$ be the truncated matrices corresponding to the top-m singular values. The reconstructed, low-frequency adjacency matrix, $\hat{\mathbf{A}}_k$, is then reconstructed as:

$$\hat{\mathbf{A}}_k = \mathbf{U}_k^{[1:m]} \boldsymbol{\Sigma}_k^{[1:m]} \mathbf{V}_k^{[1:m]^T}. \tag{4}$$

This matrix $\hat{\mathbf{A}}_k$ serves as a low-pass filtered representation of the original graph structure, preserving its most salient connectivity patterns while discarding finer, potentially disruptive information.

**Graph Spectral Structural Alignment.** This decomposition process yields two structural views for each client k: the original graph, $\mathbf{G}_k$, defined by $\mathbf{A}_k$, and the structural backbone view, $\hat{\mathbf{G}}_k$, defined by $\hat{\mathbf{A}}_k$. We then pass both views through our GNN encoder to obtain two sets of node representations. Correspondingly, let $\mathbf{h}_k$ denote the embeddings from the original graph $\mathbf{G}_k$, and $\hat{\mathbf{h}}_k$ denote the embeddings from the reconstructed graph $\hat{\mathbf{G}}_k$.

We treat these two embeddings as empirical distributions over the embedding manifold and align their global geometry using Optimal Transport (OT). Traditional alignment methods fail to capture cross-dimensional similarities, leading to the underutilization of dimensional information. To address this issue, we propose a Sinkhorn distance-based embeddings alignment method, which deploys the discrete Sinkhorn distance to comprehensively measure the distributional differences between both embeddings. We first define a ground cost matrix $\mathbf{C} \in \mathbb{R}^{N_k \times N_k}$, where $\mathbf{C}_{ij}$ is the cost of matching node $v_i$ from the original view to node $v_j$ from the backbone view. Specifically, in our implementation, A natural choice is the cosine distance:

$$\mathbf{C}_k^{ij} = 1 - \text{sim}(\mathbf{h}_k^i, \hat{\mathbf{h}}_k^j). \tag{5}$$

The SFA loss is then formulated as the Sinkhorn distance, a computationally efficient and differentiable approximation of the OT distance. It seeks a transport plan $\mathbf{P} \in \mathbb{R}^{N_k \times N_k}$ that minimizes the

total transportation cost while satisfying marginal constraints. The loss is defined as:

$$\mathcal{L}_{\text{SFA}}^{k} = \text{Sinkhorn}(\mathbf{H}_k, \hat{\mathbf{H}}_k) = \min_{\mathbf{P} \in \Pi(\mathbf{r}, \mathbf{c})} \langle \mathbf{P}_k, \mathbf{C}_k \rangle_F - \epsilon H(\mathbf{P}_k) \tag{6}$$

where $\langle \cdot, \cdot \rangle_F$ is the Frobenius dot product, which calculates the total transportation cost. It is computed by summing the element-wise products of the transport plan $\mathbf{P}$ and the cost matrix $\mathbf{C}_k$. Specifically, $\langle \mathbf{P}_k, \mathbf{C}_k \rangle_F = \sum_{i,j} \mathbf{P}_{ij} \mathbf{C}_{ij}$. $\Pi(\mathbf{r}, \mathbf{c})$ is the set of joint probability matrices with uniform marginals $\mathbf{r}$ and $\mathbf{c}$, $H(\mathbf{P}) = -\sum_{i,j} \mathbf{P}_{ij} \log \mathbf{P}_{ij}$ is the Shannon entropy of the transport plan, and $\epsilon > 0$ is the entropic regularization parameter.

### 3.2 SEMANTIC-GUIDED CONSENSUS DISTILLATION (SCD)

**Motivation.** Even when clients can learn representations resilient to local spectral and noise variations, the server still confronts significant aggregation conflicts. These conflicts arise when updates for the same conceptual class from different clients are mutually contradictory. To mitigate this issue, we propose Semantic-guided Consensus Distillation (SCD), which constructs a stable, high-quality global prototype graph to serve as a "semantic anchor" for alignment across all clients.

**Node Contribution Value.** To effectively filter core nodes within a knowledge graph and optimize the client-side knowledge extraction process, we propose a contribution assessment method based on the semantic influence of nodes. This method comprehensively considers both the Breadth Influence and Depth Influence of a node to more thoroughly characterize its importance within the knowledge network. We define the contribution of a node from the following two dimensions:

> **Depth Influence:** This metric aims to measure the core capability of a node to be strongly influenced by its local context. It is defined as the maximum influence value exerted on the node by any of its adjacent nodes, reflecting the strongest associative relationship it possesses. A node with high Depth Influence signifies that it has an exceptionally close semantic connection with at least one or more other nodes.
>
> **Breadth Influence:** This metric is designed to measure the centrality of a node as an information convergence point within the network. It is defined as the cumulative sum of influence exerted by all other nodes in the network onto the target node, reflecting the extensiveness of the influence it receives. A node with high Breadth Influence typically serves as an intersection point for multiple knowledge paths.

The calculation process primarily involves the construction of an influence matrix, the quantification of breadth and depth influences, and the final fusion to compute the contribution score. The influence matrix is calculated based on a random walk model, which can capture the indirect influence transmitted between nodes through multi-step paths. This model simulates the flow of information along the graph's edges. First, we construct the random walk transition matrix $P$ of the graph:

$$P = D^{-1}A \tag{7}$$

where $A$ is the adjacency matrix of the knowledge graph, and $D$ is the degree matrix, a diagonal matrix whose elements are the degrees of the corresponding nodes.

Building on this, we introduce a mechanism similar to Personalized PageRank to compute a global influence matrix $M$. This matrix not only considers direct connections but also incorporates multi-step reachability relationships weighted by a decay factor:

$$M = \alpha \left( I - (1 - \alpha)P \right)^{-1} \tag{8}$$

where $I$ is the identity matrix, and $\alpha \in (0, 1)$ is a smoothing factor used to balance the probabilities between the random walk and a global jump. An element $M_{i,j}$ in matrix $M$ represents the influence from node $j$ on node $i$, quantifying how much node $j$ contributes to the representation learning of node $i$ through all possible paths in the graph. For any given node $i$, $Score_i^b$ is calculated as follows:

$$Score_i^b = \frac{\sum_{j \in \mathcal{V}^{train}} M_{j,i}}{Z} \tag{9}$$

where $\mathcal{V}_L$ represents the set of all nodes in the graph, and the numerator sums the influence from all nodes on node $i$. $Z$ is a normalization factor. The Depth Influence, $Score_i^d$, is calculated as:

$$Score_i^d = \max_{j \in \mathcal{N}(i)} (M_{j,i} \cdot \tau_j) \tag{10}$$

where $\mathcal{N}(i)$ is the set of neighboring nodes of node $i$, and $\tau_j$ is an optional weight or type coefficient for node $j$, used to adjust the importance of different nodes. This formula captures the strongest determinative signal influencing node $i$ from its immediate neighborhood. Finally, we perform a linear weighted sum of the node's breadth and depth influences to obtain its final contribution score:

$$Score_i = Score_i^b + \beta \cdot Score_i^d \tag{11}$$

where $\beta$ is a hyperparameter used to adjust the weight of the Depth Influence in the total contribution score. By calculating $Score_i$ for each node, we can quantify its importance in the graph and subsequently set a threshold to filter out nodes with low influence.

**Global Knowledge Fusion.** We posit that the nodes with the highest scores constitute the Semantic Core of each client's training set, serving as the most distinct and unambiguous representatives of their respective classes. Therefore, we select only the nodes ranking in the top-1/3 based on their scores from each client $\mathcal{V}_k$ (denoted as $\mathcal{V}_k^*$) to generate local prototypes. For each class $c$, the local prototype $\mathbf{p}_k^c$ for client k is computed as follows:

$$\mathbf{p}_k^c = \frac{1}{|\mathcal{V}_k^{*c}|} \sum_{u \in \mathcal{V}_k^{*c}} \mathbf{h}_u \tag{12}$$

where $\mathcal{V}_k^{*c}$ is the set of semantic core nodes belonging to class $c$ on client $k$, and $\mathbf{h}_u$ is the embedding of node $u$. Upon completion of the local computations, each client uploads its generated prototypes and the corresponding sample counts per class to the central server. The server's objective is not to naively average them, but rather to identify and fuse the prototypes that exhibit the highest consensus and are most representative of the global distribution, in order to synthesize Global Anchors.

The server first assesses the mutual consistency among the local prototypes within a randomly selected subset of clients $\mathcal{N}_t^c$ (controlled by a participation rate $\alpha$). For each class $c$, the server constructs a prototype similarity matrix $M_{sim}^c$, where each element is defined by cosine similarity:

$$\mathbf{M}_{\text{sim}}^{ij} = \text{sim}(\mathbf{p}_i^c, \mathbf{p}_j^c) = \frac{(\mathbf{p}_i^c)^T \mathbf{p}_j^c}{|\mathbf{p}_i^c||\mathbf{p}_j^c|}, \quad \forall i, j \in \mathcal{N}_t^c \tag{13}$$

Based on this similarity matrix, the server computes a Global Cohesion Score $\mathbf{S}_i^c$ for each local prototype. This score quantifies the degree of agreement between a given local prototype and all others. A high-scoring prototype signifies that it has garnered consensus from a majority of other clients, making it more likely to reside at the center of the global class distribution.

$$S_i^c = \sum_{j \in \mathcal{N}_t^c, j \neq i} \mathbf{M}_{\text{sim}}^{ij} \tag{14}$$

Finally, for each class $c$, the server refrains from blindly averaging all prototypes. Instead, it selects only the top-K local prototypes with the highest Global Cohesion Scores for aggregation, thereby synthesizing the final Global Anchor prototype $\mathbf{P}_g^c$.

**Local Knowledge Alignment.** To guide local training, we implement a federated knowledge distillation scheme once the global model and the anchor prototype graph ($\mathbf{P}_g^c$) are disseminated to the clients. In this framework, the received global model acts as a "teacher," providing semantic guidance, while the local model acts as a "student." The knowledge transfer focuses on relational understanding rather than direct feature mimicry. Specifically, we compel the student model to replicate the teacher's perception of how a node's features relate to the global prototypes.

This is achieved by aligning their respective similarity distributions. For each node $u$, we generate a target distribution by first obtaining its feature representation $\mathbf{h}_u^g$ from the frozen teacher model. We then compute the cosine similarities between $\mathbf{h}u^g$ and all prototypes in $\mathbf{P}g$ using a function $\varphi$, and normalize these scores into a probability distribution via the softmax function $\sigma$. The student

Table 1: Performance comparison to state-of-the-art methods when poisoning $50\%$ of the full set. We report accuracy (%) and F1-macro (%) (with red/green markers indicating regression/improvement over FedAvg). The best and second-best results are highlighted with **Blue** and underline, respectively. Additional experimental results on more settings can be found in Appendix G.

| Dataset | | CORA (Acc ↑) | | | CITESEET (Acc↑) | | | PUBMED (Acc ↑) | |
|---|---|---|---|---|---|---|---|---|---|
| Noise Type | Normal | Uniform | Pair | Normal | Uniform | Pair | Normal | Uniform | Pair |
| **Normal** | | | | | | | | | |
| FedAvg [ASTAT17] | $74.50_{\downarrow00.00}$ | $48.17_{\downarrow00.00}$ | $37.20_{\downarrow00.00}$ | $64.15_{\downarrow00.00}$ | $32.59_{\downarrow00.00}$ | $34.22_{\downarrow00.00}$ | $84.16_{\downarrow00.00}$ | $78.31_{\downarrow00.00}$ | $58.53_{\downarrow00.00\uparrow}$ |
| FedProx [MLSys20] | $75.32_{\downarrow00.82}$ | $48.26_{\downarrow00.09}$ | $33.73_{\downarrow03.47}$ | $65.19_{\uparrow01.04}$ | $33.04_{\downarrow00.45}$ | $34.81_{\uparrow00.59}$ | $84.24_{\uparrow00.08}$ | $78.47_{\uparrow00.16}$ | $58.32_{\downarrow00.21}$ |
| FedDC [CVPR22] | $80.90_{\uparrow06.40}$ | $46.25_{\downarrow01.92}$ | $38.85_{\uparrow01.65}$ | $67.85_{\uparrow03.70}$ | $38.22_{\uparrow05.63}$ | $37.04_{\uparrow02.82}$ | $82.91_{\downarrow01.35}$ | $40.18_{\downarrow38.13}$ | $50.73_{\downarrow07.80}$ |
| FedDyn [ICLR21] | $79.98_{\uparrow05.48}$ | $44.42_{\downarrow03.75}$ | $38.48_{\uparrow01.20}$ | $69.04_{\uparrow04.89}$ | $34.96_{\uparrow02.37}$ | $35.11_{\uparrow00.89}$ | $81.83_{\downarrow02.33}$ | $74.77_{\downarrow03.54}$ | $51.90_{\downarrow06.63}$ |
| Scaffold [ICML20] | $\underline{81.90}_{\uparrow07.40}$ | $56.12_{\uparrow07.95}$ | $\underline{43.42}_{\uparrow06.22}$ | $\underline{70.22}_{\uparrow06.07}$ | $40.44_{\uparrow07.85}$ | $37.93_{\uparrow03.71}$ | $75.25_{\downarrow08.91}$ | $64.02_{\downarrow14.29}$ | $52.56_{\downarrow05.97}$ |
| FedGTA [VLDB24] | $72.21_{\downarrow02.29}$ | $51.83_{\uparrow03.66}$ | $36.56_{\downarrow00.64}$ | $65.33_{\uparrow01.18}$ | $37.33_{\uparrow04.74}$ | $36.15_{\uparrow01.93}$ | $82.41_{\uparrow01.75}$ | $79.20_{\uparrow00.89}$ | $57.24_{\downarrow01.29}$ |
| FedTAD [IJCAI24] | $64.90_{\downarrow09.60}$ | $46.89_{\downarrow01.28}$ | $38.76_{\uparrow01.56}$ | $64.30_{\uparrow00.15}$ | $32.44_{\downarrow00.15}$ | $34.81_{\uparrow00.59}$ | $84.36_{\uparrow00.20}$ | $77.23_{\downarrow01.08}$ | $61.51_{\uparrow02.98}$ |
| FGSSL [IJCAI23] | $68.74_{\downarrow06.16}$ | $\underline{57.40}_{\uparrow09.23}$ | $34.28_{\downarrow02.92}$ | $70.07_{\uparrow05.92}$ | $48.59_{\uparrow16.00}$ | $\underline{39.41}_{\downarrow05.19}$ | $67.03_{\downarrow17.13}$ | $66.40_{\downarrow11.91}$ | $56.43_{\downarrow02.10}$ |
| **Robust** | | | | | | | | | |
| Coteaching [NeurIPS18] | $72.03_{\downarrow02.47}$ | $45.43_{\downarrow02.74}$ | $39.03_{\uparrow01.83}$ | $64.30_{\downarrow00.15}$ | $41.48_{\uparrow08.89}$ | $35.85_{\uparrow01.63}$ | $84.26_{\uparrow00.10}$ | $76.85_{\downarrow01.46}$ | $60.02_{\uparrow01.49}$ |
| FedNoRo [IJCAI23] | $74.50_{\downarrow00.00}$ | $42.34_{\downarrow05.83}$ | $34.96_{\downarrow02.24}$ | $67.76_{\uparrow03.61}$ | $30.37_{\downarrow02.22}$ | $32.99_{\downarrow01.23}$ | $84.16_{\downarrow00.00}$ | $77.03_{\downarrow01.28}$ | $44.97_{\downarrow13.56}$ |
| FedNed [AAAI24] | $69.84_{\downarrow04.66}$ | $35.37_{\downarrow12.80}$ | $34.46_{\downarrow02.74}$ | $57.78_{\downarrow06.37}$ | $29.63_{\downarrow02.96}$ | $31.11_{\downarrow03.11}$ | $84.56_{\uparrow00.40}$ | $62.32_{\downarrow15.99}$ | $53.09_{\downarrow05.44}$ |
| FedCorr [CVPR22] | $75.14_{\uparrow00.64}$ | $52.01_{\uparrow03.84}$ | $36.20_{\downarrow01.00}$ | $38.96_{\downarrow25.19}$ | $26.07_{\downarrow06.52}$ | $29.04_{\downarrow05.18}$ | $\underline{85.93}_{\uparrow01.77}$ | $68.93_{\downarrow09.38}$ | $48.91_{\downarrow09.62}$ |
| CRGNN [NN24] | $72.30_{\downarrow02.20}$ | $49.36_{\uparrow01.19}$ | $38.03_{\uparrow00.83}$ | $63.78_{\downarrow00.37}$ | $33.78_{\uparrow01.19}$ | $38.81_{\uparrow04.59}$ | $84.08_{\downarrow00.08}$ | $40.11_{\downarrow38.20}$ | $59.72_{\uparrow01.19}$ |
| RTGNN [WWW23] | $71.46_{\downarrow03.04}$ | $46.62_{\downarrow01.55}$ | $35.28_{\downarrow00.83}$ | $67.85_{\uparrow03.70}$ | $\underline{51.11}_{\uparrow18.52}$ | $37.19_{\uparrow02.97}$ | $60.10_{\downarrow24.06}$ | $68.09_{\downarrow10.22}$ | $47.01_{\downarrow11.52}$ |
| CLNode [WSDM23] | $69.65_{\downarrow04.85}$ | $36.56_{\downarrow11.61}$ | $35.47_{\downarrow01.73}$ | $59.11_{\downarrow05.04}$ | $28.59_{\downarrow04.00}$ | $30.37_{\downarrow03.85}$ | $84.99_{\uparrow00.83}$ | $\underline{79.63}_{\uparrow01.32}$ | $53.39_{\downarrow05.14}$ |
| **AURA** | $\mathbf{85.22}_{\uparrow10.72}$ | $\mathbf{61.64}_{\uparrow13.47}$ | $\mathbf{46.27}_{\uparrow09.07}$ | $\mathbf{77.23}_{\uparrow13.08}$ | $\mathbf{52.38}_{\uparrow19.79}$ | $\mathbf{46.28}_{\uparrow12.06}$ | $\mathbf{85.95}_{\uparrow01.79}$ | $\mathbf{80.44}_{\uparrow02.13}$ | $\underline{57.53}_{\downarrow01.00}$ |
| Whole Dataset | | $48.04_{\pm0.9}$ | | | $36.94_{\pm0.3}$ | | | $69.49_{\pm0.5}$ | |

| Dataset | | COAUTHOR-PHYSICS (F1-macro ↑) | | | MINESWEEPER (F1-macro↑) | | | COAUTHOR-CS (F1-macro ↑) | |
|---|---|---|---|---|---|---|---|---|---|
| Noise Type | Normal | Uniform | Pair | Normal | Uniform | Pair | Normal | Uniform | Pair |
| **Normal** | | | | | | | | | |
| FedAvg [ASTAT17] | $24.18_{\downarrow00.00}$ | $13.76_{\downarrow00.00}$ | $27.10_{\downarrow00.00}$ | $53.51_{\downarrow00.00}$ | $33.40_{\downarrow00.00}$ | $19.38_{\downarrow00.00}$ | $02.47_{\downarrow00.00}$ | $02.47_{\downarrow00.00}$ | $02.47_{\downarrow00.00}$ |
| FedProx [MLSys20] | $23.22_{\downarrow00.96}$ | $17.50_{\uparrow03.74}$ | $25.55_{\downarrow01.55}$ | $53.30_{\downarrow00.10}$ | $33.30_{\downarrow00.10}$ | $19.35_{\downarrow00.03}$ | $02.76_{\uparrow00.29}$ | $02.45_{\downarrow00.02}$ | $03.76_{\uparrow01.29}$ |
| FedDC [CVPR22] | $13.55_{\downarrow10.63}$ | $13.46_{\downarrow00.30}$ | $14.45_{\downarrow12.65}$ | $53.59_{\downarrow00.08}$ | $35.51_{\uparrow02.11}$ | $16.60_{\downarrow02.78}$ | $03.22_{\uparrow00.75}$ | $01.90_{\downarrow00.57}$ | $03.43_{\uparrow00.96}$ |
| FedDyn [ICLR21] | $15.78_{\downarrow08.40}$ | $15.51_{\uparrow01.75}$ | $27.32_{\uparrow00.22}$ | $51.22_{\downarrow02.29}$ | $37.70_{\uparrow04.30}$ | $16.23_{\downarrow03.15}$ | $06.38_{\uparrow03.91}$ | $02.47_{\uparrow00.00}$ | $02.47_{\uparrow00.00}$ |
| Scaffold [ICML20] | $42.61_{\uparrow18.43}$ | $13.46_{\downarrow00.30}$ | $05.13_{\downarrow21.97}$ | $44.25_{\downarrow09.26}$ | $41.46_{\uparrow08.06}$ | $18.99_{\downarrow08.11}$ | $02.46_{\downarrow00.01}$ | $02.42_{\downarrow00.05}$ | $02.76_{\uparrow00.29}$ |
| FedGTA [VLDB24] | $13.72_{\downarrow10.46}$ | $17.48_{\uparrow03.72}$ | $16.24_{\downarrow10.86}$ | $53.58_{\uparrow00.07}$ | $33.19_{\downarrow00.21}$ | $21.43_{\uparrow05.67}$ | $01.35_{\downarrow01.12}$ | $03.17_{\uparrow00.70}$ | $02.59_{\uparrow00.12}$ |
| FedTAD [IJCAI24] | $22.13_{\downarrow02.05}$ | $15.42_{\uparrow01.66}$ | $08.80_{\downarrow18.30}$ | $54.11_{\uparrow00.60}$ | $32.35_{\downarrow01.05}$ | $20.12_{\uparrow06.98}$ | $02.47_{\uparrow00.00}$ | $02.56_{\uparrow00.09}$ | $06.68_{\uparrow04.21}$ |
| FGSSL [IJCAI23] | $28.13_{\uparrow03.95}$ | $13.58_{\downarrow00.18}$ | $05.06_{\downarrow22.04}$ | $44.46_{\downarrow09.05}$ | $43.50_{\uparrow10.10}$ | $16.23_{\downarrow10.87}$ | $02.89_{\uparrow00.42}$ | $02.47_{\uparrow00.00}$ | $00.31_{\downarrow02.16}$ |
| **Robust** | | | | | | | | | |
| Coteaching [NeurIPS18] | $13.46_{\downarrow10.72}$ | $14.12_{\downarrow00.36}$ | $34.85_{\uparrow07.75}$ | $44.25_{\downarrow09.26}$ | $36.77_{\uparrow03.37}$ | $17.12_{\downarrow09.98}$ | $08.00_{\uparrow05.53}$ | $07.20_{\uparrow04.73}$ | $04.36_{\uparrow01.89}$ |
| FedNoRo [IJCAI23] | $22.46_{\downarrow01.72}$ | $13.96_{\uparrow00.20}$ | $27.10_{\uparrow00.00}$ | $53.54_{\uparrow00.03}$ | $33.40_{\downarrow00.00}$ | $17.87_{\downarrow09.23}$ | $02.49_{\uparrow00.02}$ | $03.01_{\uparrow00.54}$ | $02.49_{\uparrow00.02}$ |
| FedNed [AAAI24] | $35.94_{\uparrow11.76}$ | $23.16_{\uparrow09.40}$ | $22.88_{\downarrow04.22}$ | $51.97_{\downarrow01.54}$ | $38.67_{\uparrow05.27}$ | $26.58_{\uparrow07.20}$ | $11.85_{\uparrow09.38}$ | $06.80_{\uparrow04.33}$ | $05.33_{\uparrow02.86}$ |
| FedCorr [CVPR22] | $27.93_{\uparrow03.75}$ | $21.47_{\uparrow07.71}$ | $29.42_{\uparrow02.32}$ | $44.14_{\downarrow09.37}$ | $18.40_{\downarrow15.00}$ | $16.12_{\downarrow10.98}$ | $01.34_{\downarrow01.13}$ | $02.47_{\uparrow00.00}$ | $02.62_{\uparrow00.15}$ |
| CRGNN [NN24] | $35.12_{\uparrow10.94}$ | $\underline{26.97}_{\uparrow13.21}$ | $\underline{35.83}_{\uparrow08.73}$ | $46.78_{\downarrow06.73}$ | $17.09_{\downarrow16.31}$ | $16.78_{\downarrow02.60}$ | $13.08_{\uparrow10.61}$ | $\underline{07.31}_{\uparrow04.84}$ | $\underline{11.09}_{\uparrow08.62}$ |
| RTGNN [WWW23] | $33.17_{\uparrow08.99}$ | $13.78_{\uparrow00.02}$ | $05.06_{\downarrow22.04}$ | $43.93_{\downarrow09.58}$ | $\underline{48.27}_{\uparrow14.87}$ | $\underline{31.20}_{\uparrow11.82}$ | $02.47_{\uparrow00.00}$ | $02.52_{\uparrow00.05}$ | $02.47_{\uparrow00.00}$ |
| CLNode [WSDM23] | $40.78_{\uparrow16.60}$ | $23.25_{\uparrow09.49}$ | $22.80_{\downarrow04.30}$ | $53.93_{\uparrow00.42}$ | $36.59_{\uparrow03.19}$ | $29.94_{\uparrow10.56}$ | $06.83_{\uparrow04.36}$ | $06.79_{\uparrow04.32}$ | $05.68_{\uparrow03.21}$ |
| **AURA** | $\mathbf{57.99}_{\uparrow33.81}$ | $\mathbf{68.87}_{\uparrow55.11}$ | $\mathbf{54.13}_{\uparrow27.03}$ | $\mathbf{55.29}_{\uparrow01.78}$ | $\mathbf{49.68}_{\uparrow15.88}$ | $\mathbf{47.12}_{\uparrow27.74}$ | $\mathbf{21.80}_{\uparrow19.33}$ | $\mathbf{22.31}_{\uparrow19.84}$ | $\mathbf{15.11}_{\uparrow12.64}$ |
| Whole Dataset | | $20.36_{\pm0.6}$ | | | $35.58_{\pm0.3}$ | | | $05.10_{\pm0.4}$ | |

model is then trained to minimize the Kullback-Leibler (KL) divergence between its own output distribution (derived from its feature $\mathbf{h}_u^i$) and the teacher's target distribution. This objective is formalized as our Federated Knowledge Distillation loss:

$$\mathcal{L}_{\text{FKD}} = \frac{1}{|\mathcal{V}_i|} \sum_{u \in \mathcal{V}_i} \text{KL}\left(\sigma\left(\varphi(\mathbf{h}_u^i, \mathbf{P}_g)\right), \sigma\left(\varphi(\mathbf{h}_u^g, \mathbf{P}_g)\right)\right), \tag{15}$$

where KL divergence penalizes deviations in the student's relational reasoning from the teacher's. Furthermore, we provide the overall workflow of our framework in Appendix D.

## 4 EXPERIMENT

In this section, we conduct extensive experiments to answer the following research questions: **(RQ1)** How does `AURA` perform compared to existing normal/ robust federated graph learning frameworks? **(RQ2)** What is the impact of each component of `AURA` on the overall performance? **(RQ3)** How sensitive is `AURA` to its key components and parameters?

### 4.1 EXPERIMENTAL SETUP

**Datasets and Benchmarks.** Adhering to (47), we evaluate the efficacy and robustness in five scenarios with different characteristics, including Cora (25), CiteSeer (9), PubMed (3), Physics (31), CS (32), and Minesweeper (2). Following (47), we adopt a 20%/20%/40% train/validation/test random split for the dataset. These datasets represent a wide range of domains and are commonly used in graph-based machine learning tasks. Detailed datasets information is provided in Appendix C.1.

**Comparison Method.** We compare `AURA` with several state-of-the-art methods in normal FGL and robust FGL: (1) **FedAvg** [ASTAT17] (26), (2) **FedProx** [MLSys20] (18), (3) **FedDC** [CVPR22] (7), (4) **FedDyn** [ICLR 21] (1), (5) **Scaffold** [ICML 20] (16), (6) **FedGTA** [VLDB24] (21), (7) **FedTAD** [IJCAI24] (47), (8) **FGSSL** [IJCAI23] (12), (9) textbfCoteaching [NeurIPS18] (10), (10) **FedNoRo** [IJCAI23] (39), (11) **FedNed** [AAAI24] (23), (12) **FedCorr** [CVPR22] (40);three Robust GraghLearning methods: (13) **CRGNN** [NN24] (20), (14) **RTGNN** [WWW23] (29), (15) **CLNode** [WSDM23] (38). Detailed descriptions of all the baselines can be found in Appendix C.2.

**Parameter Configuration.** The number of the SVD parameter $m$ in Equation (4) is set 10, and the number of smoothing factor $\alpha$ in Equation (8) is set 0.85. The number of the contribution score $\beta$ in Equation (11) is set 1. The detailed ablation study on hyperparameters is placed in Sec. 4.4.

## 4.2 Main Results (RQ1)

We systematically evaluated our method across a spectrum of noise environments. The experiments included both uniform and pair noise types, with noise ratios set at 0.2, 0.5 and 0.7 to simulate varying levels of label corruption. The performance results are presented in Tab. 1, demonstrating consistent superiority of our approach. We summarize the key observations as follows:

**Takeaway ❶: `AURA` demonstrates consistent robustness under diverse noise conditions.** As evidenced by Tab. 1, our method demonstrates remarkable robustness under 50% moderate-noise environments, consistently outperforming state-of-the-art FL and FGL baselines across both uniform and pair noise patterns on diverse datasets. A notable example is the Coauthor-CS dataset with GCN backbone under 50%-uniform noise, where `AURA` achieves an accuracy of 22.31%—surpassing the best baseline, CRGNN (11.09%), by a significant margin of 11.22 percentage points. This substantial performance gap underscores the effectiveness of our noise mitigation strategy.

The superiority of `AURA` extends beyond moderate-noise scenarios, as shown in Figure 3a, where it maintains an average performance gain ranging from 4.07% to 19.01% across both high-noise and low-noise environments compared to all baselines. More experimental results can be found in **??**.

Moreover, `AURA` demonstrates stable and superior performance across varying client scales, highlighting its robust scalability. As illustrated in Figure 3b, our framework achieves significantly better performance than other methods in scenarios with both 10 and 15 clients.

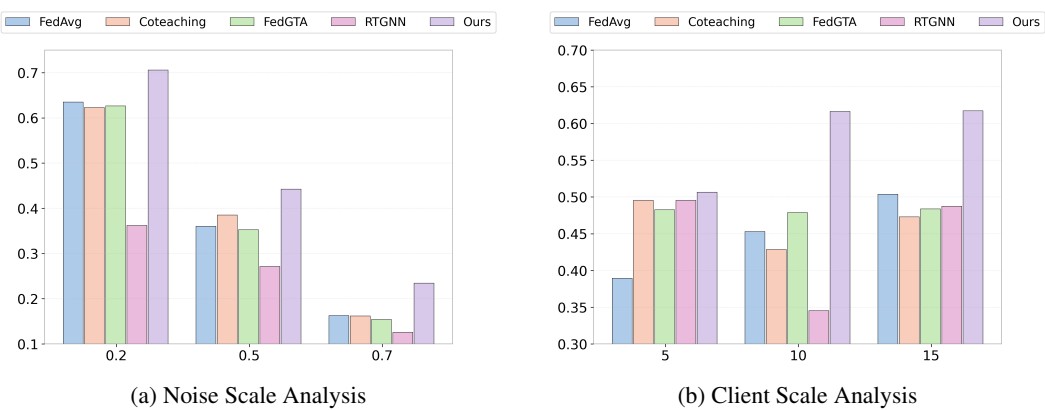

(a) Noise Scale Analysis          (b) Client Scale Analysis

Figure 3: Performance comparison under varying scales: (a) noise rate scaling; (b) number of clients scaling. Purple bars indicate the performance of `AURA`. Experiments conducted on Cora and Citeseer datasets with 50% uniform noise. Our method demonstrates consistent superiority across both scaling scenarios, exhibiting remarkable robustness to increasing noise levels and stable performance with growing client populations.

**Takeaway ❷: `AURA` achieves state-of-the-art performance in clean settings.** More importantly, Tab. 1 reveals that `AURA` not only excels in challenging noisy conditions but also achieves state-of-the-art performance in clean settings, demonstrating its versatility and general applicability across various data quality scenarios. The consistent excellence across this spectrum of conditions highlights the methodological robustness of our approach.

### 4.3 ABLATION STUDY (RQ2)

We conduct an ablation study on the key components of `AURA` both at the client-side and server-side. Tab. 2 reports the performance of `AURA` and its variants by removing specific components from SFA to SCD. We observe that both components significantly improve framework performance. Notably, `AURA` accurately decouples label noise and interference from spectral heterogeneity while ensuring excellent performance. SFA effectively filters out high-frequency variations arising from structural differences among clients, thereby mitigating client drift. Meanwhile, SCD elevates knowledge transfer from a hard alignment to a soft one, significantly enhancing generalization and stability. Importantly, when both SFA and SCD are combined, performance reaches its peak.

### 4.4 SENSITIVITY STUDY (RQ3)

We investigate the impact of final loss parameter $\alpha$ and $\beta$, on the performance of `AURA`. Specifically, we vary $\alpha \in \{0.15, 0.2, 0.25, 0.3\}$, and $\beta \in \{0.45, 0.5, 0.55\}$ on Cora and Citeseer datasets. Noise setting is 50%-uniform and 50%-pair. We observe from Figure 4 (a, b) that $\alpha = 0.25$ and $\beta = 0.5$ lead to an outstanding performance. Overall, the evaluation metrics demonstrate minimal fluctuations across different hyperparameter choices. In most of our experiments, we set the default

Table 2: **Ablation study on key components** for `AURA` on Cora and Citeseer under uniform and pair noise settings. Please see details in Sec. 4.3

| Methods | Cora-uni | Cora-pair | Cite-uni | Cite-pair |
|---------|----------|-----------|----------|-----------|
| Ours | **0.6164** | **0.4423** | **0.4989** | **0.4460** |
| w/o SFA | 0.5762 | 0.4338 | 0.4375 | 0.4260 |
| w/o SCD | 0.5555 | 0.4227 | 0.3852 | 0.3710 |
| w/o ALL | 0.4832 | 0.372 | 0.3259 | 0.3422 |

values as 0.5 for $\alpha$ and 0.25 for $\beta$. Similarly, we investigate the impact of the random walk parameter on the framework's performance. We report the performance of `AURA` on the Cora, Citeseer, and CS datasets under different noise settings, as illustrated in the Figure 4 (c, d). Based on the experimental results, we ultimately set this parameter to 0.85.

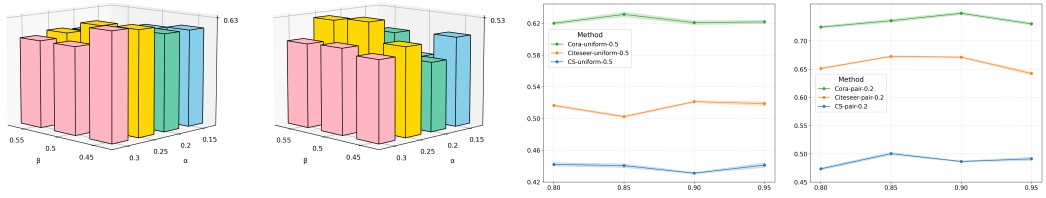

Figure 4: **Hyperparameter Analysis of `AURA`. (a,b)** Accuracy vs. $\alpha$ and $\beta$ on Cora and Citeseer under uniform-50% noise. **(c,d)** Accuracy vs. random walk parameter $\delta$ on multiple datasets: Cora (green), CS (blue), and Citeseer (yellow); (c) uniform-50% and (d) pair-20% noise.

## 5 CONCLUSION

In this work, we propose an innovative exploration of robust FGL in noisy environment. To achieve this goal, we introduce `AURA`, a novel framework that tackles this challenge from both structural and semantic perspectives. At the client level, our Structural-aware Frequency Alignment (SFA) module leverages low-rank approximation to decompose local graphs into a structural backbone and a redundant view. By enforcing consistency between these views, we effectively purify node representations from both intra-client label noise and inter-client spectral heterogeneity, mitigating client drift. Concurrently, at the server level, our Semantic-guided Consensus Distillation (SCD) method constructs a high-quality global prototype graph from consensual client knowledge. This graph serves as a semantic anchor, guiding local models through relational knowledge distillation to align their semantic understanding, which overcomes the limitations of naive aggregation. Extensive experiments on six datasets confirm the superiority of `AURA`, demonstrating significant performance gains over state-of-the-art methods under various noise conditions.

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

## A  NOTATIONS

We present a comprehensive review of the commonly used notations and their definitions in Tab. 3.

Table 3: Notations and Definitions.

| Notation | Definition |
|---|---|
| $\mathcal{G}_k = \{\mathcal{V}_k, \mathcal{A}_k, \mathcal{X}_k\}$ | Input data for the $m$-th client |
| $\mathcal{A}_k$ | The adjacency matrix of $\mathcal{G}_k$ |
| $\mathcal{X}_k$ | The feature matrix of $\mathcal{G}_k$ |
| $\mathcal{Y}_k$ | The one-hot label matrix of $\mathcal{G}_k$ |
| $h_k$ | The embeddings from the original graph $\mathcal{G}_k$ |
| $K$ | The number of clients |
| $d$ | The dimension of the node feature |
| $\epsilon$ | The fixed probability that the true label is flipped to a specific paired class |
| $\sum_k^{[1:m]}$ | The top-$m$ diagonal structural pattern for the $m$-th client |
| $\hat{\mathbf{A}}_k$ | The loss-pass filtered representation of the original graph structure |
| $C_{ij}$ | The cost of matching node $v_i$ from the original view to node $v_j$ from the backbone view |
| $P_{ij}$ | The transport plan that minimizes the total transportation cost |
| $\langle \cdot, \cdot \rangle_F$ | The Frobenius dot product |
| $\Pi(\mathbf{r}, \mathbf{c})$ | The set of joint probability matrices with uniform marginals $\mathbf{r}$ and $\mathbf{c}$ |
| $D$ | The degree matrix |
| $I$ | The identity matrix |
| $M$ | The global influence matrix |
| $\alpha$ | The smoothing factor |
| $Score_i^b$ | The Breadth influence for node $i$ |
| $Score_i^d$ | The Depth influence for node $i$ |
| $\tau_i$ | The optional weight or type coefficient for node $i$ |
| $\beta$ | The hyperparameter used to adjust the weight of Depth Influence |
| $\mathbf{p}_k^c$ | The local prototype of class $c$ for client $k$ |
| $S_i^c$ | The similarity score of prototypes i from class c |

## B  RELATED WORK.

**Federated Graph Learning(FGL).** Federated Graph Learning (FGL) extends Federated Learning (FL) to graph-structured data, enabling decentralized training while preventing the exposure of raw graph data, thus enhancing privacy protection (11; 14; 22; 4; 37; 36). Despite extensive research dedicated to enhancing model performance, the presence of noise in datasets, such as erroneous node labels, poses a significant threat to a model's generalization capabilities. To overcome this limitation, we propose a robust Federated Graph Learning (FGL) framework based on hyperspherical representation learning. This framework enhances the model's ability to extract the principal update direction from clients, thereby maintaining stable classification performance in scenarios characterized by both label noise and data heterogeneity.

**Robust Federated Learning and Graph Learning.**

The presence of noisy labels in Federated Learning, particularly when applied to graph-structured data, presents a core challenge to model robustness and generalization. Existing solutions to this problem predominantly follow two main paradigms: label correction and self-supervised learning. Label correction methods aim to directly identify and rectify erroneous annotations. Such techniques typically rely on representations learned from training data, for instance, by reassigning noisy labels based on nearest-neighbor relationships (35)in the embedding space or through predictions from a

global model(5). Although established techniques like loss correction (30; 28) and sample selection (10; 15; 17; 24; 42) exist for learning with noisy labels in traditional graph learning, they are not directly applicable to training GNNs in a federated setting, especially under conditions of limited label noise and data heterogeneity. A critical bottleneck is that most existing approaches implicitly assume a homogeneous data distribution, causing their performance to degrade in real-world non-IID scenarios. This often results in the intermixture of effective signals and noise components within the feature space. Pivoting from these limitations, our research is the first to systematically apply the self-supervised learning paradigm to construct a robust federated graph learning framework. We aim to leverage the intrinsic structure of the data itself to overcome the dual challenges posed by label noise and data heterogeneity.

## C EXPERIMENTAL DETAILS.

### C.1 DATASET DETAILS

To assess the effectiveness of `AURA`, we conduct experiments on five real-world graph datasets: Cora, CiteSeer, PubMed, Physics, and CS. Each dataset is split into training, validation, and test sets in a fixed 20%/40%/40% ratio. The key statistics of these datasets are summarized in Tab. 4. A detailed description is provided below:

- **Cora, CiteSeer, and PubMed.** These three citation network datasets are standard benchmarks in graph-based machine learning, especially for tasks like node classification and link prediction. In these datasets, nodes correspond to academic papers, while edges represent citation links. Each node is assigned a class label, and its feature vector is constructed from textual information such as words in the title or abstract. These datasets exhibit sparsity and high dimensionality, making them well-suited for evaluating the effectiveness and scalability of graph neural networks (GNNs).
- **Coauthor-Physics.** Coauthor-Physics is an academic network containing co-authorship relationships based on the Microsoft Academic Graph. Nodes in the graph represent authors and edges represent co-authorship relationships. In the dataset, authors are categorized into five classes based on their research areas, and the nodes are characterized as bag-of-words representations of keywords of papers.
- **Coauthor-CS.** Coauthor-CS represents a co-authorship network in the field of computer science, where nodes correspond to research papers, and edges denote co-authorship relations. Each paper is associated with a topic category, and features are extracted from the paper's title and abstract. This dataset is commonly used to evaluate node classification and community detection algorithms.

| Dataset | #Nodes | #Edges | #Classes | #Features |
|---|---|---|---|---|
| Cora | 2,708 | 5,278 | 7 | 1,433 |
| Citeseer | 3,327 | 4,552 | 6 | 3,703 |
| Pubmed | 19,717 | 44,324 | 3 | 500 |
| Coauthor-Physics | 34,493 | 530,417 | 5 | 8,415 |
| Coauthor-CS | 18,333 | 327,576 | 15 | 6,805 |

Table 4: **Statistics** of datasets used in experiments.

### C.2 COUNTERPART DETAILS

This section provides a comprehensive overview of the baseline approaches employed in our study.

- **FedAvg** [ASTAT17]. A foundational algorithm in Federated Learning, FedAvg operates by allowing clients to independently train models on their local datasets and subsequently transmit their model updates to a central server. The server performs a weighted aggregation of these updates to refine the global model, which is then redistributed to the clients for further local training. By transmitting only model parameters instead of raw data, FedAvg reduces communication costs and enhances privacy. However, it struggles with performance degradation in scenarios where client data distributions are highly non-IID (19; 27).

- **FedProx** [MLSys20]. As an enhancement of FedAvg, FedProx is specifically designed to address the challenges posed by statistical heterogeneity in federated learning. It introduces an additional regularization term that constrains local updates, preventing excessive divergence from the global model. This proximal term mitigates the impact of local data distribution shifts, leading to more stable convergence. By ensuring consistency in updates across clients, FedProx demonstrates improved robustness in non-IID settings.
- **FedDC** [NeurIPS20]. FedDC addresses local model drift caused by non-independent and identically distributed (Non-IID) data across clients. It introduces a local drift decoupling and correction mechanism, where an auxiliary variable on each client tracks the gap between local model updates and the global model. This approach enforces consistency at the parameter level, leading to accelerated convergence and strong performance in heterogeneous environments with partial client participation.
- **FedDyn** [ICLR21]. FedDyn addresses the inconsistency between local client objectives and the global optimization goal. It introduces a dynamic regularization method that modifies each client's objective function every round, aligning the stationary points of the local objectives with that of the global objective. This ensures that even with heterogeneous data, the local updates correctly contribute to the global goal, achieving robust performance in both convex and non-convex settings with imbalanced data and device heterogeneity.
- **Saffold** [ICML20]. Scaffold addresses the "client drift" problem in federated learning caused by heterogeneous data distributions across clients. It corrects for this drift by using control variates to estimate the update direction of the global model and the drift of each local model. This variance reduction technique modifies local updates to better align with the global objective, ensuring faster and more stable convergence while achieving higher accuracy in Non-IID settings.
- **FedGTA** [VLDB24]. FedGTA is tailored for large-scale graph federated learning, tackling issues of slow convergence and suboptimal scalability. Unlike prior methods that focus on either optimization strategies or complex local models, FedGTA integrates topology-aware local smoothing with mixed neighbor feature aggregation to improve learning efficiency (46). By leveraging graph structures in aggregation, it enhances scalability and performance in federated graph learning.
- **FedTAD** [IJCAI24]. FedTAD addresses subgraph heterogeneity in FL by decomposing local graph variations into label and structural differences, preventing inconsistent model aggregation. It enhances knowledge transfer via topology-aware distillation, boosting FL reliability and efficiency.
- **FGSSL** [IJCAI23]. FGSSL addresses local client distortion caused by both node-level semantics and graph-level structures. It improves discrimination by contrasting nodes from different classes, aligning local nodes with their global counterparts of the same class while pushing them away from different classes. To handle structural information, it transforms adjacency relationships into similarity distributions and distills relational knowledge from the global model into local models. This approach preserves both structural integrity and discriminability, achieving superior performance on multiple graph datasets.
- **Coteaching** [NeurIPS18]. Coteaching addresses the challenge of training deep learning models with noisy labels that can degrade performance. It proposes a dual-network approach where two models are trained simultaneously. In each mini-batch of data, each network selects samples with the smallest training loss—which are likely to be correctly labeled—and teaches these samples to its peer network for the subsequent parameterparameter update. This collaborative process prevents the models from overfitting to mislabeled data, significantly improving their robustness and generalization on datasets with noisy labels.
- **FedNoRo** [IJCAI23]. FedNoRo adopts a two-stage framework to address class-imbalanced global data with heterogeneous label noise in federated learning. The method first identifies noisy clients through per-class loss indicators and Gaussian Mixture Modeling, then performs noise-robust federated updates via joint knowledge distillation and distance-aware aggregation, specifically designed for realistic medical scenarios with data imbalance and complex noise patterns.
- **FedNed** [AAAI24]. FedNed adopts a negative distillation framework to effectively leverage extremely noisy clients in federated learning. The method first identifies noisy clients, then innovatively utilizes them as 'bad teachers' through a dual-training approach: one model trained on original noisy labels for reverse knowledge distillation, and another on global model-generated pseudo-labels for conditional participation in aggregation. This approach transforms noisy clients from detrimental elements into valuable contributors while progressively enhancing their trustworthiness through pseudo-label refinement

- **FedCorr** [CVPR22]. FedCorr adopts a multi-stage framework to address heterogeneous label noise in federated learning while preserving data privacy. The method first dynamically identifies noisy clients through model prediction subspace analysis and per-sample loss evaluation, then employs an adaptive local proximal regularization to handle data heterogeneity. After fine-tuning on clean clients and correcting labels for noisy ones, FedCorr performs final training across all clients to fully utilize available data, effectively handling varying noise levels without requiring prior assumptions about client noise models.
- **CRGNN** [NN24]. CRGNN addresses label noise in GNNs by combining neighborhood-based label correction and contrastive learning. It utilizes message passing neural networks to update node representations, integrating graph contrastive learning for consistent representations across augmented graph views. Finally, CGNN employs an MLP for prediction distributions and iteratively corrects noisy labels by comparing them with their neighbors and choosing the most labels.
- **RTGNN** [WWW23]. RTGNN proposes a noise governance framework that combines self-reinforcement supervision for noisy label correction and consistency regularization to prevent overfitting. The method categorizes labels into clean and noisy types, then applies adaptive supervision by rectifying inaccurate labels and generating pseudo-labels for unlabeled nodes, enabling effective learning from clean labels while mitigating noise impact.
- **CLNode** [WSDM23]. CLNode addresses the difficulty of training Graph Neural Networks (GNNs) on complex graphs for node classification tasks. It introduces a curriculum learning strategy that organizes training samples in a meaningful order, from easy to difficult, based on node characteristics. By allowing the model to first learn from simpler nodes and gradually progress to more complex ones, it helps the model better capture intricate structural patterns and relationships. This approach enhances the overall training process and leads to superior performance on various node classification benchmarks.

## D  ALGORITHM WORKFLOW

The `AURA` algorithm framework is presented in Algorithm 1.

---

**Algorithm 1 `AURA` Framework**

---

**Input:** *Communication rounds $T$, participant scale $K$, $k$-th client private model $\theta_k$, $k$-th client local data $\mathcal{G}_k$ and loss weight $\beta$*

**Output:** *The final global model $\theta_{global}$*

**for** $t = 1, 2, \cdots, T$ **do**

  *Client Side:* **for** $m = 1$ *to* $K$ *in parallel* **do**

    $\hat{\mathbf{A}}_k \leftarrow$ SVDandReconstruct$(\mathcal{G}_k, \mathcal{A}_k)$by Equation (4)  // Decompose original graph and reconstruct low-frequency adjacency matrix

    $\mathcal{L}_{SFA} \leftarrow$ Sinkhorn$(\mathbf{H}_k, \hat{\mathbf{H}}_k)$by Equation (6)  // Graph Spectral Structural Alighment

    $Score_i^b \leftarrow$ BreadthInfluence$(\mathcal{G}_k)$by Equation (9)  // Calculate Breadth Influence for each node

    $Score_i^d \leftarrow$ DepthInfluence$(\mathcal{G}_k)$by Equation (10)  // Calculate Depth Influence for each node

    $\mathbf{P} \leftarrow$ CalculatePrototypes$(Score_i^b, Score_i^d, \mathcal{G}_k)$by Equation (12)  // Select node and calculate prototypes

    $\mathcal{L}_{FKD} \leftarrow$ LocalKnowledgeAlignment$(\mathbf{P}_g, \mathbf{h}^g, \mathbf{h}^i)$by Equation (15)  // Semantic-guided Consensus Distillation

    $\theta_m^{t+1} \leftarrow$ LocalUpdating$(\theta_m^t, \mathcal{L}_{SFA} + \beta\mathcal{L}_{FKD})$  // Backward propagation

  *Server Side:*

  $\mathcal{M}_{sim}^c =$ CalculateSimilarity$(\mathbf{p}^c), \forall c$ by Equation (13) // Calculate prototypes similarity

  $\mathbf{P}_g^c \leftarrow$ SelectPrototypes$(\mathcal{M}_{sim}^c), \forall c$ by Equation (14) // Select Prototypes by similarity matrix

  $\theta_{global} \leftarrow$ Aggregate$(\theta_k), \forall k$  // Clean clients aggregation

  $\theta_k \leftarrow \theta_{global}, \forall k$ // Distribute parameters to clients

**return** $\theta_{global}$

---

Table 5: Performance comparison to state-of-the-art methods when poisoning $20\%$ of the full set.

| Dataset | CORA (Acc ↑) | | | CITESEET (Acc↑) | | | PUBMED (Acc ↑) | | |
|---|---|---|---|---|---|---|---|---|---|
| Noise Type | Normal | Uniform | Pair | Normal | Uniform | Pair | Normal | Uniform | Pair |
| **Normal** | | | | | | | | | |
| FedAvg [ASTAT17] | $74.50_{\uparrow00.00}$ | $68.83_{\uparrow00.00}$ | $67.55_{\uparrow00.00}$ | $64.15_{\uparrow00.00}$ | $43.70_{\uparrow00.00}$ | $52.44_{\uparrow00.00}$ | $84.16_{\uparrow00.00}$ | $84.36_{\uparrow00.00}$ | $83.07_{\uparrow00.00}$ |
| FedProx [MLSys20] | $75.32_{\uparrow00.82}$ | $68.92_{\uparrow00.09}$ | $69.56_{\uparrow02.01}$ | $65.19_{\uparrow01.04}$ | $45.33_{\uparrow01.63}$ | $51.26_{\downarrow01.18}$ | $84.24_{\uparrow00.08}$ | $84.46_{\uparrow00.10}$ | $83.25_{\uparrow00.18}$ |
| FedDC [CVPR22] | $80.90_{\uparrow06.40}$ | $67.55_{\downarrow01.28}$ | $70.02_{\uparrow02.47}$ | $67.85_{\uparrow03.70}$ | $53.04_{\uparrow09.34}$ | $57.33_{\uparrow04.89}$ | $82.91_{\downarrow01.35}$ | $53.04_{\downarrow31.32}$ | $44.81_{\downarrow38.26}$ |
| FedDyn [ICLR21] | $79.98_{\uparrow05.48}$ | $67.82_{\downarrow01.01}$ | $69.20_{\uparrow01.65}$ | $69.04_{\uparrow04.89}$ | $52.89_{\uparrow09.19}$ | $56.59_{\uparrow04.15}$ | $81.83_{\downarrow02.33}$ | $81.10_{\downarrow01.97}$ | $75.81_{\downarrow07.26}$ |
| Scaffold [ICML20] | $\underline{81.90}_{\uparrow07.40}$ | $73.40_{\uparrow04.57}$ | $\underline{71.85}_{\uparrow04.30}$ | $70.22_{\uparrow06.07}$ | $56.96_{\uparrow13.26}$ | $62.07_{\uparrow09.63}$ | $75.25_{\downarrow08.91}$ | $72.77_{\downarrow11.59}$ | $65.54_{\downarrow17.53}$ |
| FedGTA [VLDB24] | $72.21_{\downarrow02.29}$ | $71.57_{\uparrow02.74}$ | $68.46_{\downarrow00.91}$ | $65.33_{\uparrow01.18}$ | $48.74_{\uparrow05.04}$ | $56.59_{\uparrow04.15}$ | $82.41_{\downarrow01.75}$ | $83.35_{\downarrow01.01}$ | $81.83_{\downarrow01.24}$ |
| FedTAD [IJCAI24] | $64.90_{\downarrow09.60}$ | $62.61_{\downarrow06.22}$ | $63.35_{\downarrow04.20}$ | $64.30_{\uparrow00.15}$ | $44.44_{\uparrow00.74}$ | $52.59_{\uparrow00.15}$ | $84.36_{\uparrow00.20}$ | $84.59_{\uparrow00.23}$ | $82.97_{\downarrow00.10}$ |
| FGSSL [IJCAI23] | $68.74_{\downarrow06.16}$ | $69.20_{\uparrow00.37}$ | $66.27_{\downarrow01.28}$ | $\underline{70.07}_{\uparrow05.92}$ | $\underline{64.30}_{\uparrow20.60}$ | $62.81_{\uparrow10.37}$ | $67.03_{\downarrow17.13}$ | $72.34_{\downarrow12.02}$ | $66.57_{\downarrow16.50}$ |
| **Robust** | | | | | | | | | |
| Coteaching [NeurIPS18] | $72.03_{\downarrow02.47}$ | $67.37_{\downarrow01.46}$ | $66.82_{\downarrow00.73}$ | $64.30_{\uparrow00.15}$ | $54.22_{\uparrow10.52}$ | $56.74_{\uparrow04.30}$ | $84.26_{\uparrow00.10}$ | $82.52_{\downarrow01.84}$ | $\underline{83.40}_{\uparrow00.33}$ |
| FedNoRo [IJCAI23] | $74.50_{\uparrow00.00}$ | $68.83_{\uparrow00.00}$ | $67.37_{\downarrow00.18}$ | $67.76_{\uparrow03.61}$ | $43.70_{\uparrow00.00}$ | $52.44_{\uparrow00.00}$ | $84.16_{\uparrow00.00}$ | $84.36_{\uparrow00.00}$ | $83.07_{\downarrow00.00}$ |
| FedNed [AAAI24] | $69.84_{\downarrow04.66}$ | $56.58_{\downarrow12.25}$ | $55.94_{\downarrow11.61}$ | $57.78_{\downarrow06.37}$ | $37.33_{\downarrow06.37}$ | $46.07_{\downarrow06.37}$ | $84.56_{\uparrow00.40}$ | $81.63_{\downarrow02.73}$ | $79.91_{\downarrow03.16}$ |
| FedCorr [CVPR22] | $75.14_{\uparrow00.64}$ | $67.82_{\downarrow01.01}$ | $65.63_{\downarrow01.92}$ | $38.96_{\downarrow25.19}$ | $30.07_{\downarrow13.63}$ | $34.07_{\downarrow18.37}$ | $85.93_{\uparrow01.77}$ | $\underline{84.69}_{\uparrow00.33}$ | $84.64_{\uparrow01.57}$ |
| CRGNN [NN24] | $72.30_{\downarrow02.20}$ | $67.28_{\downarrow01.55}$ | $67.92_{\uparrow00.37}$ | $69.33_{\uparrow05.18}$ | $60.00_{\uparrow16.30}$ | $62.37_{\uparrow09.93}$ | $84.08_{\downarrow00.08}$ | $83.40_{\downarrow00.96}$ | $83.38_{\uparrow00.31}$ |
| RTGNN [WWW23] | $71.46_{\downarrow03.04}$ | $59.60_{\downarrow09.23}$ | $54.57_{\downarrow12.98}$ | $67.85_{\uparrow03.70}$ | $60.33_{\uparrow16.63}$ | $66.67_{\uparrow14.23}$ | $60.10_{\downarrow24.06}$ | $74.49_{\downarrow09.87}$ | $65.71_{\downarrow17.36}$ |
| CLNode [WSDM23] | $69.65_{\downarrow04.85}$ | $57.68_{\downarrow11.15}$ | $57.59_{\downarrow09.96}$ | $59.11_{\downarrow05.04}$ | $40.59_{\downarrow03.11}$ | $48.74_{\downarrow03.70}$ | $84.99_{\uparrow00.83}$ | $82.06_{\downarrow02.30}$ | $80.74_{\downarrow02.33}$ |
| **AURA** | $\mathbf{85.22}_{\uparrow10.72}$ | $\mathbf{75.50}_{\uparrow06.67}$ | $\mathbf{73.58}_{\uparrow06.03}$ | $\mathbf{77.23}_{\uparrow13.08}$ | $\mathbf{65.56}_{\uparrow21.86}$ | $\mathbf{67.26}_{\uparrow14.82}$ | $\mathbf{85.95}_{\uparrow01.79}$ | $\mathbf{86.15}_{\uparrow01.79}$ | $\mathbf{82.11}_{\downarrow00.96}$ |
| Whole Dataset | $62.13_{\pm0.9}$ | | | $49.12_{\pm0.0}$ | | | $80.54_{\pm0.0}$ | | |

| Dataset | COAUTHOR-PHYSICS (F1-macro ↑) | | | MINESWEEPER (F1-macro↑) | | | COAUTHOR-CS (F1-macro ↑) | | |
|---|---|---|---|---|---|---|---|---|---|
| Noise Type | Normal | Uniform | Pair | Normal | Uniform | Pair | Normal | Uniform | Pair |
| **Normal** | | | | | | | | | |
| FedAvg [ASTAT17] | $24.18_{\uparrow00.00}$ | $15.58_{\uparrow00.00}$ | $37.93_{\uparrow00.00}$ | $53.51_{\uparrow00.00}$ | $50.16_{\uparrow00.00}$ | $47.22_{\uparrow00.00}$ | $02.47_{\uparrow00.00}$ | $02.47_{\uparrow00.00}$ | $02.47_{\uparrow00.00}$ |
| FedProx [MLSys20] | $23.22_{\downarrow00.96}$ | $19.81_{\uparrow04.23}$ | $\underline{46.06}_{\uparrow08.13}$ | $53.54_{\uparrow00.03}$ | $51.25_{\uparrow01.09}$ | $47.42_{\uparrow00.20}$ | $02.76_{\uparrow00.29}$ | $02.72_{\uparrow00.25}$ | $03.12_{\uparrow00.65}$ |
| FedDC [CVPR22] | $13.55_{\downarrow10.63}$ | $13.46_{\downarrow02.12}$ | $13.46_{\downarrow24.47}$ | $53.59_{\uparrow00.08}$ | $46.52_{\downarrow03.64}$ | $44.33_{\downarrow02.89}$ | $03.22_{\uparrow00.75}$ | $03.09_{\uparrow00.62}$ | $02.54_{\uparrow00.07}$ |
| FedDyn [ICLR21] | $15.78_{\downarrow08.40}$ | $13.61_{\downarrow01.97}$ | $21.69_{\downarrow16.24}$ | $51.22_{\downarrow02.29}$ | $50.23_{\uparrow00.07}$ | $44.33_{\downarrow02.89}$ | $06.38_{\uparrow03.91}$ | $06.05_{\uparrow03.58}$ | $04.09_{\uparrow01.62}$ |
| Scaffold [ICML20] | $\underline{42.61}_{\uparrow18.43}$ | $15.21_{\downarrow00.37}$ | $24.66_{\downarrow13.27}$ | $44.25_{\downarrow09.26}$ | $46.45_{\downarrow03.71}$ | $45.01_{\downarrow02.21}$ | $02.46_{\downarrow00.01}$ | $02.47_{\uparrow00.00}$ | $02.47_{\uparrow00.00}$ |
| FedGTA [VLDB24] | $13.72_{\downarrow10.46}$ | $13.59_{\downarrow01.99}$ | $32.84_{\downarrow05.09}$ | $53.58_{\uparrow00.07}$ | $51.40_{\uparrow01.24}$ | $46.13_{\uparrow01.09}$ | $01.35_{\downarrow01.12}$ | $03.11_{\uparrow00.64}$ | $03.54_{\uparrow01.07}$ |
| FedTAD [IJCAI24] | $22.13_{\downarrow02.05}$ | $13.46_{\downarrow02.12}$ | $13.75_{\downarrow24.18}$ | $54.11_{\uparrow00.60}$ | $51.85_{\uparrow01.69}$ | $46.13_{\uparrow01.09}$ | $02.46_{\downarrow00.00}$ | $02.78_{\uparrow00.31}$ | $02.47_{\uparrow00.00}$ |
| FGSSL [IJCAI23] | $28.13_{\uparrow03.95}$ | $13.46_{\downarrow02.12}$ | $34.25_{\downarrow03.68}$ | $44.46_{\downarrow09.05}$ | $41.98_{\downarrow08.18}$ | $44.33_{\downarrow02.89}$ | $02.83_{\uparrow00.36}$ | $02.47_{\uparrow00.00}$ | $02.14_{\downarrow00.33}$ |
| **Robust** | | | | | | | | | |
| Coteaching [NeurIPS18] | $13.46_{\downarrow10.72}$ | $32.29_{\uparrow16.71}$ | $42.52_{\uparrow04.59}$ | $44.25_{\downarrow09.26}$ | $44.78_{\downarrow05.38}$ | $44.33_{\downarrow02.89}$ | $08.00_{\uparrow05.53}$ | $07.37_{\uparrow04.90}$ | $08.22_{\uparrow05.75}$ |
| FedNoRo [IJCAI23] | $22.46_{\downarrow01.72}$ | $15.58_{\uparrow00.00}$ | $37.93_{\uparrow00.00}$ | $53.54_{\uparrow00.03}$ | $51.25_{\uparrow01.09}$ | $47.42_{\uparrow00.20}$ | $02.49_{\uparrow00.02}$ | $02.42_{\uparrow00.05}$ | $02.49_{\uparrow00.02}$ |
| FedNed [AAAI24] | $35.94_{\uparrow11.76}$ | $41.90_{\uparrow26.32}$ | $38.51_{\uparrow00.58}$ | $51.97_{\downarrow01.54}$ | $51.94_{\uparrow01.78}$ | $\underline{48.79}_{\uparrow01.57}$ | $11.85_{\uparrow09.38}$ | $11.57_{\uparrow09.10}$ | $06.75_{\uparrow04.28}$ |
| FedCorr [CVPR22] | $27.93_{\uparrow03.75}$ | $28.45_{\uparrow15.87}$ | $34.12_{\downarrow03.81}$ | $44.14_{\downarrow09.37}$ | $41.12_{\downarrow09.04}$ | $44.12_{\downarrow03.10}$ | $01.34_{\downarrow01.13}$ | $08.06_{\uparrow05.59}$ | $02.96_{\uparrow00.49}$ |
| CRGNN [NN24] | $35.12_{\uparrow10.94}$ | $\underline{47.44}_{\uparrow31.86}$ | $44.14_{\uparrow06.21}$ | $46.78_{\downarrow06.73}$ | $43.12_{\downarrow07.04}$ | $44.33_{\downarrow02.89}$ | $\underline{13.08}_{\uparrow10.61}$ | $\underline{11.39}_{\uparrow08.92}$ | $\underline{11.70}_{\uparrow09.23}$ |
| RTGNN [WWW23] | $33.17_{\uparrow08.99}$ | $13.46_{\downarrow02.12}$ | $40.28_{\uparrow02.35}$ | $43.93_{\uparrow09.58}$ | $44.25_{\uparrow05.91}$ | $44.31_{\uparrow02.91}$ | $02.47_{\uparrow00.00}$ | $09.78_{\uparrow07.31}$ | $02.47_{\uparrow00.00}$ |
| CLNode [WSDM23] | $40.78_{\uparrow16.60}$ | $37.96_{\uparrow22.38}$ | $28.14_{\downarrow09.79}$ | $\underline{53.93}_{\uparrow00.42}$ | $\underline{52.41}_{\uparrow02.25}$ | $48.30_{\uparrow01.08}$ | $06.83_{\uparrow04.36}$ | $06.83_{\uparrow04.36}$ | $06.81_{\uparrow04.34}$ |
| **AURA** | $\mathbf{57.99}_{\uparrow33.81}$ | $\mathbf{66.64}_{\uparrow51.06}$ | $\mathbf{56.53}_{\uparrow18.60}$ | $\mathbf{55.29}_{\uparrow01.78}$ | $\mathbf{54.34}_{\uparrow04.18}$ | $\mathbf{54.34}_{\uparrow07.12}$ | $\mathbf{21.80}_{\uparrow19.33}$ | $\mathbf{21.56}_{\uparrow19.84}$ | $\mathbf{20.17}_{\uparrow17.70}$ |
| Whole Dataset | $25.45_{\pm0.9}$ | | | $42.46_{\pm0.3}$ | | | $07.46_{\pm0.0}$ | | |

# E  ETHICS AND SOCIETAL IMPACT

This work aims to advance the robustness of FGL by addressing challenges such as label noise and data heterogeneity. While the research primarily focuses on improving model performance and generalization in decentralized environments, we acknowledge the broader implications of deploying such models in real-world settings. One key ethical concern lies in the use of sensitive data in FL environments. Although FL inherently protects privacy by keeping data local, the potential for adversarial manipulation of models or the inadvertent leakage of sensitive information remains a risk. In this study, we do not introduce any new sensitive datasets or personal data into the models, and we adhere to the principles of privacy-preserving machine learning.

# F  THE USE OF LARGE LANGUAGE MODEL (LLMs)

Our use of Large Language Models (LLMs) was strictly limited to polishing the language ancearch and intellectual content of this papegenerating figures for the manuscript. All underlying reincluding the **AURA** frawork, its theoretical foundations, experimental design, and thanalysis of results, was completed entirely by the authors without assistance from LLMs.

# G  ADDITIONAL EXPERIMENTAL RESULTS

We place additional Acc and F1-macro score results under 0.2 noisy label ratios in Tab. 5.

