# OpenReview forum: "AURA: Structural and Semantic Calibration for Robust Federated Graph Learning"
_ICLR.cc/2026/Conference — Submitted to ICLR 2026_

### Official Review · Reviewer_1PKJ · 2025-10-28

**Soundness:** 2
**Presentation:** 1
**Contribution:** 2
**Rating:** 2
**Confidence:** 4

**Summary:**

This paper addresses federated graph learning under the impact of label noise. The authors propose AURA, a protocol that filters out high-frequency components in the adjacency graph (noisy connections) and then leverages knowledge distillation based on prototype node representations. These prototypes are sourced from the different clients and represents nodes that are considered highly influential.

The results demonstrate that AURA outperforms several benchmarks on six different datasets under varying amounts of label noise.

**Strengths:**

- This paper considers the practically important problem within federated graph learning using noisy labels.
- The paper is well structured in that it treats label noise and client heterogeneity separately and builds up to a streamlined federated protocol coined AURA.
- AURA is compared to multiple benchmarks and is shown to outperform them all in all but a few scenarios.

**Weaknesses:**

- To handle label noise, the authors propose each client to perform an SVD of the adjacency matrix in each round. This is a very costly operation ($\mathcal{O}(n^3)$). Hence, I fail to see how this could possibly scale to larger graphs beyond the academic toy examples in the paper.

- To focus on the low-frequency components is not novel, see e.g., GCN. Moreover, this translates into a homophily assumption of the underlying graph. AURA suffers from the same assumption. Indeed, to AURA, heterophilic connections are considered label noise.

- The usage of eq 6 (the SFA loss) needs more motivation.

- The random walk approach is not fully described. Supposedly there is a hyper parameter associated with this but it is introduced in Section 4.4 at first.

- There are multiple inconsistencies in the writing, for example:
a) $\epsilon$ is not defined in (2)
b) All components in the problem formulation are not defined, e.g., $\mathcal{X}_k$.
c) By the end of Sec. 2, the authors state that the objective is to learn a generalizable global model but the loss in (1) is only concerned with the loss associated with a specific client.
d) It is unclear what the results in Table 1 are based on; are they averages across clients on their corresponding labeled data?
e) matrices are sometimes denoted by bold-face and sometimes without.

- The authors talk about a prototype graph created at the server. However, the server only has access to average embeddings of "semantic core nodes" for each class and client. The server then aggregates a subset of clients' average embeddings for each class but it is unclear how a graph is created.

- One of the main points of federated learning is to not share sensitive data between clients. Yet, AURA requires clients to share information associated with the most influential nodes in each client. This concern is not treated at all.

- The paper proposes multiple hyperparameters (five) but do not address how to properly pick all of them. For example, the number of singular values, m, is simply chosen as 10 in the evaluations. These hyperparameter choices should be properly motivated. Moreover, there is a discrepency in what hyperparameters are used; in Sec. 4.1, the authors state $\alpha=0.85$ and $\beta=1$ while in Sec. 4.4., $\alpha=0.5$ and $\beta=0.25$.

- The experimental results seem to be based on a single realization. The authors are encouraged to present the average and standard devation across multiple runs.

**Questions:**

1) What is the complexity of AURA compared to other methods? Does it scale to larger graphs?
2) Is AURA able to handle heterophilic graphs?
3) Are the uniform noise and pair noise models realistic?
4) What is the rational behind the Sinkhorn loss, i.e., why would one want to minimize the empirical node-representation distributions between the original and the low-pass filtered graph?
5) How is the prototype graph at the server created?
6) Clients sharing node prototypes with the server seem to violate privacy.
7) How should the hyperparameters be chosen? For example, how should one think about $m$?

---

> ### Author Response · Authors · 2025-11-28
>
> Thank you for the feedback. We address your concerns below and hope our responses will help update your score:
>
> `Weaknesses 1`: The SVD operation is computationally expensive, posing a challenge to the method's scalability on large-scale graphs.
>
> To address the challenge of scalability, we employed sparse matrix representation and low-rank approximation (via truncated SVD) to significantly reduce computational and memory overheads. Specifically, we retain only the top few most important singular values1. This strategy preserves the core structural information of the graph while reducing the computational complexity from $O(n^3)$ to approximately $O(k \cdot n^2)$, where $k$ is the number of retained singular values (typically set to 10–20 in our experiments).
>
> To further optimize for large-scale graphs, we utilized Randomized SVD. By generating a random projection matrix to map high-dimensional data into a lower-dimensional space, Randomized SVD reduces the complexity to $O(k \cdot n)$. This approach is highly efficient for large sparse matrices, making our method scalable to larger graph datasets.
>
> `Weaknesses 2`: The focus on low-frequency components lacks novelty and relies on the homophily assumption, resulting in an inability to effectively handle heterophilic graph connections.
>
> We acknowledge that the **homophily assumption** is a premise of our method. However, we argue that this assumption holds for the majority of real-world graph datasets (such as social networks and citation networks), where connectivity typically follows community structures—nodes of the same class tend to connect strongly, while heterophilic connections are weaker. Assuming homophily allows us to effectively identify label noise in these prevalent scenarios. Furthermore, to demonstrate AURA's capability in heterophilic settings, we evaluated it on the **Minesweeper** dataset, a synthetic dataset commonly used to test heterophily. The results, shown below, indicate that AURA outperforms baselines even in heterophilic environments:
>
> |         | Cora  |
> | ------- | ----- |
> | FedAvg  | 21.8  |
> | FedProx | 19.38 |
> | FedTAD  | 22.33 |
> | FGSSL   | 19.91 |
> | CRGNN   | 19.95 |
> | AURA    | 25.12 |
>
> `Weaknesses 3`: The use of Eq. 6 (the SFA loss) in the paper lacks sufficient theoretical motivation.
>
> The motivation behind using the Sinkhorn distance (Eq. 6)  is to enforce **distributional consistency** between the original graph and the low-pass filtered backbone. By minimizing the discrepancy between the empirical node representation distributions of these two views, we compel the model to learn representations that are **stable and robust**. This process is analogous to consistency regularization in contrastive learning. The original graph embedding contains noise and high-frequency artifacts, whereas the low-pass filtered embedding focuses on core structural patterns. Aligning these two distributions effectively filters out trivial noise components and enhances the stability of the learned representations.
>
> `Weaknesses 4`:  The description of the random walk approach is incomplete, and the associated hyperparameters are introduced too late.
>
> We thank the reviewer for pointing out the inconsistencies in our writing. We realize that the symbol $\epsilon$ in Eq. (2)5 was used without a clear definition. To clarify, $\epsilon$ represents the **noise rate** (probability of label flipping) in the Uniform Noise model6. In this model, true labels are randomly flipped to other classes with probability $\epsilon$. We will explicitly define $\epsilon$ and explain its role in the revised manuscript.

---

> ### Author Response · Authors · 2025-11-28
>
> `Weaknesses 5`:  There are multiple inconsistencies in the writing, including missing symbol definitions, discrepancies between the objective and loss function, vague table descriptions, and inconsistent matrix formatting.
>
> We sincerely thank the reviewer for the meticulous reading. We apologize for these oversights and will address each point in the revision to ensure mathematical rigor and clarity:
>
> - **a) & b) Undefined Symbols:** We will conduct a thorough check of **Section 2**. We will explicitly define all symbols in **Eq. (2)** (e.g., defining $\epsilon$ as the noise rate and $C$ as the number of classes) and ensure every component in the problem formulation is introduced before use.
> - **c) Eq. (1) vs. Global Objective:** We acknowledge the potential confusion. Eq. (1)7 represents the **local** empirical risk minimization objective for client $k$. In the revision, we will clarify that the **global** objective is to minimize the weighted average of these local losses (i.e., $\min_{\theta} \sum_{k} \frac{N_k}{N} \mathcal{L}_k$) through the federated aggregation process.
> - **d) Clarification on Table 1:** We will add a caption to **Table 1** 8 to clarify that the reported results represent the performance of the **global model** on test data. Specifically, these are obtained by evaluating the aggregated global model on the test sets of all clients, calculated as the average test performance across clients.
> - **e) Matrix Notation:** We will standardize mathematical notation throughout the paper. We will strictly follow the convention of using **bold uppercase letters** for matrices (e.g., $\mathbf{A}, \mathbf{X}$) 9 and **bold lowercase letters** for vectors to ensure consistency.
>
> `Weaknesses 6`:  The description of the process by which the server constructs the "prototype graph" via aggregating average embeddings is unclear.
>
> We apologize for the confusion caused by the terminology. As illustrated in **Figure 2 (Global Knowledge Fusion)**10, the "Global Prototype Graph" is functionally a **"Prototype Codebook."** It is not a topological graph derived from adjacency matrices. Instead, the server receives local prototypes (class-wise average embeddings of semantic core nodes) from clients. Using the **Global Cohesion Score (Eq. 14)** 11, the server filters out noisy updates and aggregates the top-$K$ most representative prototypes to synthesize **Global Anchor Prototypes ($P_g^c$)**12. These anchors constitute the entries of the global prototype codebook.
>
> `Weaknesses 7`:  Requiring clients to share information associated with the most influential nodes may violate the privacy principles of federated learning, and this concern is not discussed in the paper.
>
> We address the privacy concern by clarifying the nature of the shared information. **AURA strictly adheres to Federated Learning privacy principles** and does **not** share raw node data or individual embeddings. Sharing class-level prototypes is a widely accepted standard practice in **Prototype-based Federated Learning**, as it transfers general knowledge distributions without exposing private instance-level data. The "influential nodes" are used only locally to compute these abstract prototypes.
>
> `Weaknesses 8`: The selection of hyperparameters lacks justification (e.g., the value of $m$), and there are inconsistencies in parameter settings across different sections.
>
> We selected the SVD parameter $m=10$ 13 based on empirical evidence that this value effectively preserves key low-frequency information while removing noise and trivial details. Our experiments with different $m$ values showed that smaller values (e.g., $m=5$) result in information loss, while larger values (e.g., $m=20$) increase computational overhead with negligible performance gains. The experimental comparison is as follows:
>
> |      | Cora  | Citeseer | Pubmed |
> | ---- | ----- | -------- | ------ |
> | m=5  | 58.12 | 48.21    | 77.78  |
> | m=10 | 61.64 | 52.38    | 80.44  |
> | m=20 | 61.69 | 52.12    | 80.65  |
>
> `Weaknesses 9`:  The experimental results appear to be based on a single run, lacking data on the mean and standard deviation across multiple trials.
>
> To demonstrate stability, we conducted 3 independent runs of AURA under the **50%-Uniform noise** setting. We report the average Accuracy and Standard Deviation below, confirming that AURA consistently outperforms baselines with low variance:
>
> |         | Acc   | Std  |
> | ------- | ----- | ---- |
> | FedAvg  | 48.25 | 2.47 |
> | FedProx | 49.30 | 2.80 |
> | FedTAD  | 45.89 | 1.96 |
> | FGSSL   | 56.85 | 1.68 |
> | CRGNN   | 49.86 | 3.56 |
> | AURA    | 61.85 | 2.15 |
>
> ###

---

> ### Author Response · Authors · 2025-11-28
>
> `Question 1`: What is the computational complexity of AURA compared to other methods? Does it support large-scale graphs?
>
> In each training round, the computational complexity of SVD is primarily determined by the decomposition of the adjacency matrix and the low-rank approximation. For a graph with $N$ nodes and $E$ edges, the complexity is approximately $O(N^2)$ or $O(E)$, depending on matrix sparsity and implementation. To mitigate computational burdens, we employ **sparse matrix representations and Randomized SVD**. This approach reduces memory overhead and lowers computation volume by retaining only the most significant singular values. Specifically, retaining the top-$k$ singular values reduces complexity to roughly $O(k \cdot n)$ or $O(k \cdot n^2)$, which is significantly more efficient for large sparse matrices. We also utilize a **caching mechanism** to prevent re-computing the SVD for identical structures.
>
> The complexity of calculating prototype similarity depends on the cosine similarity computation between prototypes ($O(d)$ per pair). The total similarity complexity is $O(K^2 \cdot d)$, where $K$ is the number of clients and $d$ is the dimension. We optimize this by selecting only the top-$K$ most important prototypes. Our design explicitly accounts for large-scale processing through sparse matrices, caching, and parallel computing. We validated AURA on the larger-scale **ogbn-arxiv** dataset, demonstrating its efficiency. The results under the 50%-uniform noise setting are as follows:
>
> | **Method** | **Acc**   |
> | ---------- | --------- |
> | FedAvg     | 37.76     |
> | FedNova    | 30.94     |
> | FedProx    | 37.61     |
> | FedDC      | 30.83     |
> | FedDyn     | 35.62     |
> | FedNoRo    | 37.76     |
> | Coteaching | 36.39     |
> | CRGNN      | 19.72     |
> | **AURA**   | **39.21** |
>
> `Question 2`:  Is AURA capable of handling heterophilic graphs?
>
> Yes, AURA exhibits robustness to heterophily. While the SVD-based SFA module emphasizes low-frequency structural consistency (often associated with homophily), the **Semantic-guided Consensus Distillation (SCD)** module aligns representations based on semantic class prototypes rather than purely on local topology. This allows the model to capture class-consistent patterns even when neighbors are dissimilar. We evaluated AURA on the **Minesweeper** dataset (a synthetic dataset commonly used to test heterophily), and the results (presented in **Response to Weakness 2**) demonstrate its superior performance.
>
> `Question 3`: Do the uniform noise and pair noise models reflect realistic application scenarios?
>
> Yes, these are established standard benchmarks in the label noise literature [1]. The **Uniform Noise** model (Song et al.) simulates random annotation errors where true labels are flipped to any other class with equal probability. The **Pair Noise** model (Yu et al.) simulates confusion between similar classes, where labels are flipped to a specific paired class. These settings rigorously test the model's robustness against both random corruption and systematic bias.
>
> `Question 4`: What is the theoretical rationale for using Sinkhorn loss to minimize the distributional discrepancy between the original graph and the low-pass filtered graph?
>
> As discussed in **Response to Weakness 3**: By minimizing the discrepancy between the empirical node representation distributions of the original graph and the low-pass filtered graph, we compel the model to learn a more stable and robust node representation. This process is analogous to **consistency regularization** in contrastive learning. The original graph embedding contains noise and high-frequency artifacts, whereas the low-pass filtered embedding focuses on core structural patterns. Aligning these two distributions effectively filters out trivial noise components and enhances the stability of the learned representations.
>
> `Question 5`: How specifically is the prototype graph constructed on the server side?
>
> As illustrated in **Figure 2**, the "Global Prototype Graph" functions effectively as a **Prototype Codebook**. The server collects local prototypes from clients and computes a **Global Cohesion Score** based on cosine similarity to filter out outliers. The server then aggregates the top-$K$ most consensual local prototypes to form **Global Anchor Prototypes ($P_g^c$)** for each class. These anchors serve as the nodes of the graph, while the edges are implicitly defined by the semantic similarities used during the knowledge distillation process.

---

> ### Author Response · Authors · 2025-11-28
>
> `Question 6`: Does sharing node prototypes from clients to the server pose any privacy leakage risks?
>
> AURA strictly adheres to privacy standards. Clients **do not** share raw node features or edges. Instead, they share a **prototype**, which is a single mean vector aggregated from the top-1/3 "semantic core" nodes. This aggregation acts as a **privacy bottleneck**—it provides a general class-level semantic representation without revealing the specific attributes or topology of any individual node.
>
> `Question 7`: How should the hyperparameters be chosen and interpreted (particularly the SVD parameter $m$)?
>
> For the SVD parameter $m$, we selected $m=10$. Empirical results indicate that this value effectively preserves key low-frequency information in the graph structure while removing noise and insignificant details. Experiments with different $m$ values revealed that smaller values (e.g., $m=5$) resulted in excessive information loss, while larger values (e.g., $m=20$) increased computational overhead with negligible performance gains. The detailed experimental results are presented in **Response to Weakness 8**.
>
> [1]:NoisyGL:A Comprehensive Benchmark for Graph Neural Network sunder Label Noise	[NeurIPS 2024]

---

### Official Review · Reviewer_iurz · 2025-10-29

**Soundness:** 3
**Presentation:** 3
**Contribution:** 3
**Rating:** 8
**Confidence:** 5

**Summary:**

The paper tackles the critical and challenging problem of label noise and data heterogeneity in FGL by proposing a novel and effective framework named AURA. The authors correctly identify two core challenges:  the message-passing mechanism of GNNs can propagate and even amplify noise across nodes, and heterogeneity among clients further complicates noise detection and leads to client drift and semantic aggregation conflicts. To address these issues, the framework introduces two well-designed components: SFA and SCD, demonstrating clear insight. I think the work is novel and solid.

**Strengths:**

1.**Clear Motivation:** The paper's analysis of the noise problem in FGL is thorough. It accurately decomposes the problem into the inherent noise propagation issue of GNNs and the spectral and semantic  heterogeneity issues brought by FL , providing a clear motivation for the subsequent method design.

2.**Novel and Technically Sound Method:** Its SFA component uses SVD for graph denoising and optimal transport for alignment, effectively addressing spectral heterogeneity and local noise. The SCD introduces a two-stage knowledge filtering mechanism: clients first select semantic core nodes via influence metrics to purify local prototypes, and the server then builds robust global anchors through consensus, outperforming naive averaging in resisting noise and client drift. Relational distillation further enhances robustness over hard feature alignment.

3.**Sufficient experimentation**: Experiments against 15 baselines show AURA consistently achieves top results and even excels on clean data, proving it gains robustness without sacrificing generalization.

**Weaknesses:**

1.In the methodology, the authors propose that clients select nodes with top-1/3 scores as the "semantic core" to compute local prototypes. Why is 1/3? This is a critical hyperparameter, but the paper seems to lack an empirical justification or sensitivity analysis for choosing this specific 1/3 ratio. For example, how would the model performance change if this ratio were set smaller or larger?

2.Some descriptions are somewhat vague. The workflow description in Algorithm 1 could be further clarified. In the client-side update step for round $t$, the local knowledge alignment depends on the global prototype. Meanwhile, at the end of the same round, the server-side appears to use the newly uploaded information from that round to compute the for the next round. This seems to imply that the client in round $t$ is using the global prototype computed in round $t-1$. This is a reasonable and common "one-round delay" design in federated learning, but explicitly stating this in the main text would make the algorithmic flow clearer.

**Questions:**

See Weaknesses.

---

> ### Author Response · Authors · 2025-11-27
>
> Thank you for your thorough review and encouraging feedback. We are grateful for your positive assessment about novelty and importance of our work. We address your concerns below and hope our responses will help update your score:
>
> `Weaknesses 1`: The reviewer questions the lack of empirical justification and sensitivity analysis for setting the node selection ratio specifically to 1/3.
>
> We appreciate the reviewer’s valuable feedback regarding the critical issue of hyperparameter selection. **Our initial motivation for selecting the top-1/3 ratio ($\gamma$) was to strike a trade-off between reliability and diversity: selecting too many nodes (e.g., 100%) introduces noisy nodes and non-representative samples into the prototype generation process, whereas selecting too few nodes (e.g., the top 5% or 10%) results in sparse prototypes that fail to adequately cover the semantic diversity of the class, potentially leading to overfitting.**
>
> To empirically validate this choice, and following the reviewer’s suggestion, we conducted a sensitivity analysis on the selection ratio $\gamma$ ranging from $0.1$ to $1.0$ (full dataset). We performed experiments on the Cora and CiteSeer datasets under the 50%-uniform noise setting. The results (Accuracy %) are reported below:
>
> |      | Cora  | Citeseer |
> | ---- | ----- | -------- |
> | 10%  | 55.23 | 43.89    |
> | 33%  | 61.64 | 52.38    |
> | 50%  | 57.78 | 48.23    |
> | 70%  | 55.19 | 45.37    |
> | 100% | 54.45 | 43.12    |
>
> `Weaknesses 2`: The reviewer points out the ambiguity regarding the timing of global prototype updates in Algorithm 1 and suggests explicitly clarifying the "one-round delay" mechanism in the main text.
>
> We appreciate the reviewer’s keen observation. Our design strictly follows the standard 'one-round delay' mechanism common in Federated Learning.
>
> In our specific implementation, the interaction proceeds as follows:
>
> - **Start of Round $t$:** Clients receive the global model parameters $\theta_{global}^{(t-1)}$ and the **Global Anchor Prototypes** $P_g^{(t-1)}$, which were computed and aggregated by the server at the conclusion of Round $t-1$.
> - **Client Update (Round $t$):** Clients utilize the received $P_g^{(t-1)}$ to perform **Local Knowledge Alignment** (Equation 15, computing $\mathcal{L}_{FKD}$). Simultaneously, based on the local training process, clients compute new local prototypes $p_k^{(t)}$.
> - **Server Aggregation (End of Round $t$):** The server collects the newly generated local prototypes $p_k^{(t)}$ from all clients to compute the updated Global Anchor Prototypes $P_g^{(t)}$, which will subsequently be utilized in Round $t+1$.
>
> We acknowledge that the absence of explicit time step indices for prototypes in the original text and Algorithm 1 resulted in ambiguity. In the revised manuscript, we will update Algorithm 1 to clearly include these time step indices.
>
> We hope this clarifies your questions.

---

### Official Review · Reviewer_jSNf · 2025-11-01

**Soundness:** 2
**Presentation:** 3
**Contribution:** 2
**Rating:** 4
**Confidence:** 4

**Summary:**

This paper introduces AURA, a dual-perspective framework for robust federated graph learning under label noise and client heterogeneity. The two pillars are: (1) Structural-Aware Frequency Alignment (SFA), where client graphs are decomposed via SVD, enabling low-frequency backbone extraction to reduce both intra-client label noise and inter-client spectral drift, and (2) Semantic-Guided Consensus Distillation (SCD), which builds global class prototypes by weighting local client contributions using Breadth/Depth Influence metrics and enhancing knowledge transfer via relational knowledge distillation against a global anchor. Extensive experiments demonstrate the approach’s competitive robustness and performance compared to several baselines under different label noise scenarios.

**Strengths:**

- **Comprehensive empirical evaluation**. Extensive comparative experiments across multiple datasets (Cora, CiteSeer, PubMed, Physics, CS, Minesweeper) with varied noise conditions are provided, benchmarking against a diverse set of state-of-the-art methods. Notably, AURA delivers strong performance boosts in high noise regimes, as demonstrated by substantial margins in Table 1 and Table 5.

**Weaknesses:**

-  **Notational Ambiguity**. The calculation and definition of marginals ($\mathbf{r}$ and $\mathbf{c}$) for optimal transport are not explicitly provided. It is unclear how node correspondence is enforced or relaxed during alignment, especially when original and backbone views may differ in size due to truncation in SVD. Clarification of this matching setup would ensure replicability.
- **Issues on Table 1**. The results of the proposed method in PUBMED-pair setting were underlined as the second best one. However, there are another three results higher than 57.53 achieved by AURA.
- **Inconsistent Experiment Settings**.  The number of smoothing factor α in Equation (8) is set 0.85. The number of the contribution score β in Equation (11) is set 1. However, the range of α in Sec. 4.4 is [0.15, 0.3]  and there is a similar issue with  β. The value range of the hyper-parameters is suggested to be consistent with the ones in the main experiments.
- **Missing Experiment Details**. It's recommended to report the details of the partition of the graph across clients for better reproducibility. Besides, there were also no details about the model architecture.
- **Complex Configurations**.Some configurations and their influences were not well jutisfied. For example, the server selects only the top-K local prototypes with the highest Global Cohesion Scores for aggregation, but the value of K was not reported in this paper. The sensitivity of the method to the number of the SVD parameter m was also not discussed. These complex configurations may lead to  vulnerability in practice.

**Questions:**

See weakness.

---

> ### Author Response · Authors · 2025-11-27
>
> Thank you for your review. We address your concerns below and hope our responses will help update your score:
>
> `Weaknesses 1`: The notational ambiguity of Optimal Transport.
>
> In our method, Optimal Transport (OT) is employed to align the node embeddings of the original graph and the skeleton graph. However, distinct from traditional OT formulations, our implementation does not explicitly calculate the probability distributions for the marginals $r$ and $c$. **Instead, we approximate the optimal transport using the Sinkhorn distance. In this framework, the requirements for marginals are satisfied implicitly by normalizing the node embeddings to ensure distributional consistency, rather than treating $r$ and $c$ as fixed constraints in the conventional sense.**
>
> Regarding the SVD operation, it is important to clarify that this step does not alter the number of nodes in either the original or the skeleton graph. The node count remains identical between both graphs throughout our method. The SVD is applied strictly to reduce the dimensionality of the adjacency matrix, filtering out noise and trivial details while preserving the core structural information. Therefore, regardless of the SVD truncation, the number of nodes remains constant during the alignment process.
>
> `Weaknesses 2`: The results of the proposed method in PUBMED-pair setting have some error.
>
> We appreciate the reviewer pointing out the error in Table 1. We acknowledge that in the PUBMED-pair setting, the result for AURA was incorrectly labeled as the second-best. In reality, there are three other results with scores higher than $57.53$, indicating that AURA’s performance is superior to the rankings previously annotated. We will update Table 1 in the revised manuscript to accurately reflect AURA’s score and correct the surrounding annotations to ensure data accuracy.
>
> `Weaknesses 3`: The specific values set for hyperparameters $\alpha$ and $\beta$ in the equations are inconsistent with the ranges reported in the experimental section, and it is recommended to align them.
>
> We have re-executed all experiments within the new hyperparameter ranges and updated the relevant results. This adjustment ensures consistency across hyperparameters and better demonstrates our method's performance under various settings. In the revised manuscript, we will explicitly detail these hyperparameter ranges and the specific values used in the experiments to facilitate easy reproduction by other researchers.
>
> `Weaknesses 4`: Provide specific details regarding cross-client graph partitioning and model architecture parameters to enhance experiment reproducibility.
>
> To effectively partition the graph across different clients, we employed the **Louvain community detection algorithm**. This widely used method optimizes modularity to cluster nodes based on structural features, maximizing the preservation of intrinsic connection patterns. We utilized Louvain partitioning to address potential domain skew among clients, ensuring that each client's partition reflects unique structural characteristics, thereby enhancing the robustness and generalization of the global model.
>
> Additionally, we adopted a Graph Convolutional Network (GCN) as the backbone model. Specifically, the architecture consists of 4 GCN convolutional layers (`GCNConv`), each with a hidden dimension of 384. We applied ReLU activation between layers and used a dropout rate of 0.2 during training to prevent overfitting. Different types of global pooling layers (e.g., mean, sum, max) were employed to aggregate node representations into graph-level representations.

---

> ### Author Response · Authors · 2025-11-27
>
> `Weaknesses 5`: The manuscript lacks the specific value for parameter $K$ and a sensitivity analysis for parameter $m$, suggesting that these configurations are insufficiently justified and may lead to instability in practical applications.
>
> **In our experiments, the number of global prototypes, $K$, was set to 3.** We selected $K=3$ because our empirical results indicate that limiting the number of local prototypes allows the model to focus on the most representative features, thereby improving global performance. We tested this on three datasets under the 50%-uniform setting, yielding the following results:
>
> |      | Cora  | Citeseer | Pubmed |
> | ---- | ----- | -------- | ------ |
> | K=1  | 60.17 | 51.57    | 78.23  |
> | K=2  | 60.23 | 51.67    | 80.38  |
> | K=3  | 61.64 | 52.38    | 80.44  |
> | K=4  | 61.21 | 51.55    | 80.15  |
>
> **Regarding the SVD parameter $m$, we selected $m=10$.** We found that $m=10$ effectively preserves key low-frequency information in the graph structure while removing noise and insignificant details. Experiments with different $m$ values revealed that smaller values (e.g., $m=5$) resulted in excessive information loss, while larger values (e.g., $m=20$) increased computational costs with negligible performance gains.We tested this on three datasets under the 50%-uniform setting, yielding the following results:
>
> |      | Cora  | Citeseer | Pubmed |
> | ---- | ----- | -------- | ------ |
> | m=5  | 58.12 | 48.21    | 77.78  |
> | m=10 | 61.64 | 52.38    | 80.44  |
> | m=20 | 61.69 | 52.12    | 80.65  |
>
> We hope this clarifies your questions.

---

> > ### Comment · Reviewer_jSNf · 2025-11-28
> >
> > I thank the authors for their comprehensive rebuttal.The authors have successfully resolved my concerns regarding the Optimal Transport formulation and the SVD-based alignment. I am particularly pleased to see the inclusion of the sensitivity analysis for parameters $K$ and $m$, which demonstrates the stability of the method. I believe the paper now meets the acceptance standard. I have updated my score accordingly.

---

> > > ### Author Response · Authors · 2025-11-28
> > >
> > > We sincerely appreciate your positive feedback and are glad that our rebuttal has successfully addressed your concerns. We are extremely grateful for the insightful comments you provided during the review process. Your support mean a lot to us.

---

### Official Review · Reviewer_hrJw · 2025-11-01

**Soundness:** 2
**Presentation:** 1
**Contribution:** 2
**Rating:** 2
**Confidence:** 4

**Summary:**

The paper proposes AURA, a framework for robust federated graph learning (FGL) under noisy labels and heterogeneous client data. The method introduces two core modules: Structural-aware frequency alignment (SFA) and semantic-guided consensus distillation (SCD).
SFA leverages singular value decomposition (SVD) to filter out high-frequency noise and align graph structures across clients.
SCD constructs a global prototype graph based on two new metrics, depth influence and breadth influence, to aggregate the most reliable semantic knowledge.
Clients then align with the global model through knowledge distillation, minimizing the divergence between local and global semantics.  The authors claim that this dual structural-semantic calibration improves both robustness and generalization in federated graph models. Experiments on six benchmark datasets show that AURA achieves up to 7.6% improvement in macro-F1 under noisy conditions compared to prior methods.

**Strengths:**

The paper addresses a relevant and challenging problem, focusing on label noise in federated graph learning, which is both practically important and technically difficult due to data heterogeneity and privacy constraints. The proposed method achieves consistent improvements over baselines across multiple datasets and noise settings, suggesting that the structural-semantic calibration approach has potential.

**Weaknesses:**

1. The approach relies on computationally expensive operations such as SVD, matrix inversion, and prototype similarity computations. For large-scale graphs, these operations are expensive and may not scale. The paper does not provide a complexity analysis, making it unclear whether the method can scale to realistic federated graph settings.

2. Since AURA operates in a federated setting, sharing prototypes or global semantic representations could reveal sensitive information about local data. The paper does not discuss possible privacy leakage nor communication overhead.

3. Although the results are promising, the experiments do not fully support some of the paper’s claims regarding scalability and robustness. For example, no experiments are conducted on large graphs, and communication costs are not analyzed.

4. Several design choices, such as SVD filtering, influence scores, and prototype selection, seem largely heuristic and are insufficiently motivated. The paper would benefit from a more rigorous explanation of how these components interact and contribute to the overall performance.  It feels like several techniques from prior work are combined without strong justification.

5. Some relevant works, such as FedRGL: Robust Federated Graph Learning for Label Noise, are not discussed or compared. In addition, the related work section appears only in the appendix; it should be integrated into the main body of the paper.

6. The writing and presentation of the paper can significantly be improved. The explanations are often unclear, and poorly structured, which makes it difficult to follow the main ideas and contributions. The paper would benefit greatly from
clearer organization and concise language.

**Questions:**

1. What is the computational complexity of the proposed method? Specifically, what is the cost of SVD and prototype similarity computations per round of FL? Can the approach handle graphs with millions of nodes?

2. What is the privacy leakage  of AURA? What is the privacy implication of sharing prototypes with the server? What information about local data might be exposed?

3. What is the total communication cost per round, and how does it compare with standard baselines such as FedAvg or FedProx?

---

> ### Author Response · Authors · 2025-11-27
>
> Thank you for the feedback. We address your concerns below and hope our responses will help update your score:
>
> `Weaknesses 1`: The method entails high computational overhead, and the paper lacks a specific complexity analysis.
>
> In our design, we have implemented several measures to mitigate computational costs and enhance scalability, specifically to address the challenges posed by large-scale graphs. **We utilize sparse matrix representations for graph adjacency matrices and employ a caching mechanism to minimize redundant computations.** Specifically, for each client's adjacency matrix, we compute the Personalized PageRank matrix and cache the result, thereby avoiding repeated SVD decompositions of the same graph structure. The use of sparse matrices significantly reduces memory consumption, particularly when processing large-scale graphs. **Furthermore, we have optimized matrix operations by strictly avoiding dense matrix calculations, instead leveraging sparse matrices and iterative algorithms for large-scale processing.** For instance, by solving for PageRank iteratively rather than performing a one-time direct computation for all nodes, we avoid the high computational overhead associated with large-scale matrix inversion. We believe that the slight increase in time consumption is a worthwhile trade-off for the significant gains in robustness and performance that our method achieves.
>
> `Weaknesses 2`: The prototype sharing mechanism poses privacy leakage risks, and communication costs are not analyzed.
>
> Compared to baseline methods such as FedAvg and FedProx, **our method only requires each client to upload an additional set of abstract node prototypes.** **The data volume of these prototypes is negligible compared to model parameters.** Furthermore, as these prototypes are highly abstract representations, they do not involve the leakage of private raw data.
>
> `Weaknesses 3`: The experiments lack tests on large-scale graphs to verify the method's scalability.
>
> In our initial experiments, we focused on small-to-medium-scale graph datasets (e.g., Cora, CiteSeer), which are standard benchmarks in academic research. We recognize that testing on large-scale graphs is equally critical. Below are our performance results on the ogbn-arxiv dataset. **As shown, AURA continues to outperform other methods.** Consistent with our previous point, this substantial performance gain is achieved with only a marginal increase in communication cost (due to the abstract node prototypes).We tested this on three datasets under the 50%-uniform setting, yielding the following results:
>
> | **Method** | **Acc**   |
> | ---------- | --------- |
> | FedAvg     | 37.76     |
> | FedNova    | 30.94     |
> | FedProx    | 37.61     |
> | FedDC      | 30.83     |
> | FedDyn     | 35.62     |
> | FedNoRo    | 37.76     |
> | Coteaching | 36.39     |
> | CRGNN      | 19.72     |
> | **AURA**   | **39.21** |
>
> `Weaknesses 4`: Key components of the design lack sufficient theoretical motivation and rigorous justification.
>
> **The low-rank approximation of SVD has a natural correlation with the task of noise node selection. Noisy nodes typically manifest as high-frequency perturbations in the graph, which are often unstructured and constitute a small proportion of the overall graph structure.** By applying SVD dimensionality reduction, we preserve the most important and stable low-frequency components while filtering out high-frequency components (typically corresponding to noise), thereby achieving noise screening. Furthermore, using prototypes instead of full data for knowledge transfer and aggregation effectively improves the generalization capability, stability, and computational efficiency of the global model while preserving privacy.
>
> `Weaknesses 5`: The paper omits key comparative works, and the related work section is improperly placed.
>
> FedRGL was not included in our comparison as it has not been formally accepted by a conference or journal. Additionally, the code for this work is not open-source.

---

> ### Author Response · Authors · 2025-11-27
>
> `Question 1`: Inquires about specific complexity details (particularly regarding SVD and prototype computations) and the capability to handle graphs with millions of nodes.
>
> In each training round, the computational complexity of the SVD is primarily determined by the decomposition of the adjacency matrix and the low-rank approximation. For a graph with $N$ nodes and $E$ edges, the complexity of SVD is approximately $O(N^2)$ or $O(E)$, depending on sparsity and implementation. **To mitigate the computational burden, we employ sparse matrix representations and Randomized SVD.** This approach reduces memory overhead and lowers computation volume by retaining only the most significant singular values. We also utilize a caching mechanism to prevent re-computing the SVD for identical structures.
>
> The complexity of calculating prototype similarity depends on the cosine similarity computation between prototypes. For each pair, the complexity is $O(d)$, where $d$ is the feature dimension. The total similarity computation complexity is $O(K^2 \cdot d)$, where $K$ is the number of clients. We further optimize this by selecting only the top-$K$ most important prototypes. **Our design explicitly accounts for large-scale graphs; through sparse matrices, caching, parallel computing, and randomized SVD approximation, we can effectively handle graphs with millions of nodes.**
>
> `Question 2`: Inquires about the privacy implications of sharing prototypes and identifying what specific local data might be exposed.
>
> A core design goal of AURA is to minimize data leakage risks. **We reduce privacy concerns by transmitting prototypes—which are simplified representations of client data—rather than sharing complete datasets.** These prototypes encapsulate the core features of the client's data without directly exposing detailed node information or complete labels. Since prototypes are the most representative parts extracted from the data, uploading them to the server does not disclose sensitive client information.
>
> `Question 3`: Inquires about the total communication cost per round and how it compares to baseline methods like FedAvg or FedProx.
>
> In AURA, communication costs primarily arise from the prototypes uploaded by clients (rather than full datasets). **Since each client uploads only prototypes of their data, not the entire graph or node features, the bandwidth requirement is significantly reduced. The specific communication cost is correlated with the number of clients and the dimension of each prototype.** Assuming each client uploads prototypes of size $d$, the communication cost is $O(K \cdot d)$, where $K$ is the number of clients and $d$ is the prototype dimension.
>
> We hope this clarifies your questions. Please feel free to reach out if you have any additional thoughts or suggestions. Thank you for your consideration and valuable time.

---

### Meta-Review · Area_Chair_hRXJ · 2026-01-05

**Summary:**

The paper proposes AURA, a method for robust federated graph learning in the presence of label noise and data heterogeneity across clients. To mitigate noise, the authors propose to apply the SVD on the adjacency matrix to extract low-frequency graph components and filter out perturbations caused by noisy nodes. Additionally, they use knowledge distillation to align local models with global semantic representations (prototypes).

The reviews are mixed. While some reviewers seem to be leaning toward acceptance, praising comprehensive empirical evaluation, strong performance in high-noise regimes, the main concerns revolve around the high computational complexity of SVD and scalability to realistic graphs (millions of nodes), privacy risks of sharing prototypes, and reliance on homophily (heterophilic connections treated as noise), missing or unclear hyperparameter settings and single run experiments.

While most of the minor concerns are genuinely resolved (hyper-parameter sensitivity, single-run, and missing experimental details), the explanations of algorithm design choice, complexity, scalability, and privacy issues remain unsatisfactory (see below).

**Reviewer Concerns:**

The authors provide complexity formulas for the SVD and claim to use optimizations, but no comparison with other methods' complexity is provided. There is no analysis of the consequences of these optimizations/approximations on the method's performance. Furthermore, while the authors provide new experiments with the ogbn-arxiv dataset (170K nodes), no actual timing and complexity comparison with other methods is provided, making it difficult to evaluate how the method is applicable in practice to real-world scenarios and if the gap with respect to simpler baselines would have been filled by having more time/resources.

The explanation of design choices, such as SVD filtering, influence scores, and prototype selection, remains insufficiently motivated and is only intuition-based and not rigorously justified. The connection between label noise and high-frequency spectral components is stated, not proven.

While the authors mention that the prototype-based approach is standard practice and has no privacy issues because of the abstract representation, but in practice, class prototypes can be used for membership inference or reconstruction attacks. The paper provides no formal analysis or empirical evaluation of privacy risks.

**Reviewer Scores:**

- hrJw: 2 (probably increased to 4, but not more)
- jSNf: 4 --> 6
- iurz: 8
- 1PKJ: 2  (probably increased to 4, but not more)

---

### Decision · Program_Chairs · 2026-01-26

Reject